# Wild-Time: A Benchmark of in-the-Wild Distribution Shift over Time

**Huaxiu Yao**[1*]**, Caroline Choi**[1*]**, Bochuan Cao**[2]**, Yoonho Lee**[1]**, Pang Wei Koh**[1]**, Chelsea Finn**[1]

[1]Stanford University, [2]Pennsylvania State University

wildtime@googlegroups.com

## Abstract

Distribution shift occurs when the test distribution differs from the training distribution, and it can considerably degrade performance of machine learning models deployed in the real world. *Temporal shifts* – distribution shifts arising from the passage of time – often occur gradually and have the additional structure of timestamp metadata. By leveraging timestamp metadata, models can potentially learn from trends in past distribution shifts and extrapolate into the future. While recent works have studied distribution shifts, temporal shifts remain underexplored. To address this gap, we curate Wild-Time, a benchmark of 5 datasets that reflect temporal distribution shifts arising in a variety of real-world applications, including patient prognosis and news classification. On these datasets, we systematically benchmark 13 prior approaches, including methods in domain generalization, continual learning, self-supervised learning, and ensemble learning. We use two evaluation strategies: evaluation with a fixed time split (Eval-Fix) and evaluation with a data stream (Eval-Stream). Eval-Fix, our primary evaluation strategy, aims to provide a simple evaluation protocol, while Eval-Stream is more realistic for certain real-world applications. Under both evaluation strategies, we observe an average performance drop of 20% from in-distribution to out-of-distribution data. Existing methods are unable to close this gap. Code is available at https://wild-time.github.io/.

## 1   Introduction

Distribution shift occurs when the test distribution differs from the training distribution. *Temporal shifts* – distribution shifts that arise from the passage of time – are a common type of distribution shift. Due to non-stationarity, production (i.e. test) data shifts over time [39]. This degrades the performance of machine learning systems deployed in the real world. For example, Lazaridou et al. [60] found that neural language models perform worse when predicting future utterances from beyond their training period, and that their performance worsens with time. As another example, flu incidence prediction from Internet search queries performed remarkably well in 2008 [27]. However, using the same model in 2013 incorrectly predicted double the

Figure 1: An illustration of temporal distribution shift on Yearbook. In Wild-Time, we split the train and test sets by timestamp and observe performance drops between train and test distributions.

---

*Huaxiu Yao and Caroline Choi contributed equally.

36th Conference on Neural Information Processing Systems (NeurIPS 2022) Track on Datasets and Benchmarks.

| Datasets | Yearbook | FMoW-Time | MIMIC-IV | | HuffPost | arXiv |
|---|---|---|---|---|---|---|
| | | | Readmission | Mortality | | |
| Input (x) | yearbook photos | satel. image | diagnosis, treatment (ICD9) | | article headline | paper title |
| Prediction (y) | gender | land use | readmission | mortality | news tag | primary category |
| Time Range | 1930 - 2013 | 2002 - 2017 | 2008 - 2019 | | 2012 - 2018 | 2007 - 2022 |
| # Examples | 37,189 | 118,886 | 270,617 | | 63,907 | 2,057,952 |
| **Train Example** | Female | Residential | Diagnosis: 560, 998, 788, 278, E878, 311, V88, V10, 266, 272  Treatment: 456, 545  Readmission: No; Mortality: No | | Killer Fail: How Romney's Broken Orca App Cost Him Thousand of Votes  TECH | The Limitations of Deep Learning in Adversarial Settings  cs.CR |
| **Test Example** | Female | Park | Diagnosis: 155, 456, 452, 572  Treatment: 423, 549, 990, 990  Readmission: Yes; Mortality: Yes | | Possible Autopilot Use Probed After Tesla Crashes at 60mph  TECH | Progressive-Scale Boundary Blackbox Attack via Projective Gradient Estimation  cs.LG |

Figure 2: The Wild-Time benchmark includes a collection of 5 datasets with 6 tasks, from [26], [48], [76], [17]. For each task, we train models on the past and evaluate it in the future. We list the input, prediction, time range and the number of examples for each task.

incidence [8]. Finally, in Figure 1, the style of yearbook portraits of American high schoolers [26] change over the decades. As a result, models trained on earlier years and evaluated on future years suffer substantial drops in performance.

Though temporal shifts are ubiquitous in real-world scenarios, they remain understudied. Prior benchmarks for out-of-distribution robustness in the wild focus on domain shifts and subpopulation shifts [55, 72, 109, 89]. Many popular benchmarks that feature a stream of data, such as those used in continual learning [1, 13, 54, 92, 111, 14, 69, 84, 95], contain a manually delineated set of tasks and artificial sequential variations, which are not representative of natural temporal shifts. These include small-image sequences with disparate label splits (e.g., Split TinyImageNet [61], Split CIFAR [57]), different kinds of image transformations to MNIST digits (e.g., Rainbow MNIST [23]), or different visual recognition targets [64] (cf. Section 6.) Recent works have investigated natural temporal distribution shifts in different domains such as drug discovery [37], visual recognition [10], and sepsis prediction [30] and created datasets in each of these domains. However, there does not exist a systematic study of real-world temporal distribution shifts and a benchmark spanning various domains. Here, we curate a collection of these datasets and create a benchmark that allows researchers to easily evaluate their methods across multiple domains.

This paper presents **Wild-Time** ("in-the-Wild distribution shifts over Time"), a benchmark of in-the-wild gradual temporal distribution shifts together with two comprehensive evaluation protocols. In Wild-Time, we investigate real-world temporal distribution shifts across a diverse set of tasks (Figure 2), including portraits classification [26], ICU patient readmission prediction [48], ICU patient mortality prediction [48], news tag classification [76], and article category classification [17]. The distribution shifts in these applications happen naturally due to the passage of time, which the datasets reflect through changing fashion and social norms [26], atmospheric conditions [72], and current events [75, 76]. We propose two evaluation strategies for Wild-Time: evaluation with a fixed time split (Eval-Fix) and with a data stream (Eval-Stream).

On these datasets, we evaluate several representative approaches in continual learning, invariant learning, self-supervised learning, and ensemble learning. We extend invariant learning methods to the temporal distribution shift setting. While prior invariant learning approaches are trained on clearly delineated sets of distributions, we consider a stream of unsegmented observations, where domain labels are not provided. To extend domain invariant methods to the temporal distribution shift setting, we construct domains to be different windows of time. More specifically, all temporal windows of a certain window size are treated as a domain, allowing us to directly apply invariant learning approaches over the constructed domains.

The main conclusions of our benchmark analysis are that invariant learning, self-supervised learning, and continual learning approaches do not show substantial improvements compared to standard ERM training. To make the Wild-Time datasets accessible for future research, we released a Python

package that automates data loading and baseline training at https://wild-time.github.io/. We hope that Wild-Time will accelerate the development of temporally robust models.

## 2   Problem and Evaluation Settings

We define the temporal robustness setting. Following [55], we view the entire data distribution as a mixture of $T$ timestamps $\mathcal{T} = \{1, \ldots, T\}$. Each timestamp $t$ is associated with a data distribution $P_t$ over $(x, y)$, where $x$ and $y$ represent input features and labels, respectively, and all examples are sampled from the data distribution $P_t$. To formulate the temporal distribution shift setting, we define the training distribution as $P^{tr} = \sum_{t=1}^{T} \lambda_t^{tr} P_t$, and the test distribution as $P^{ts} = \sum_{t=1}^{T} \lambda_t^{ts} P_t$. Note that, here, timestamp differs from the notion of "domain" used in other works on distribution shift [24, 63, 3, 2, 55, 108]. In the temporal shift setting, we do not require distribution shift between consecutive timestamps, i.e., we can have $P_t = P_{t-1}$. Based on the problem setting, we will detail the criteria to select datasets and the evaluation strategies in Wild-Time.

### 2.1   Criteria for Dataset Selection

In Wild-Time, we select datasets using three criteria:

- **Naturally Occurring Temporal Shifts.** We select real-world datasets that consist of data collected over time and contain timestamp metadata. We select datasets for which it is natural to train on the past and test into the future, and we include datasets from a diverse collection of domains, including vision, healthcare, and language modeling.

- **Temporal Distribution Shifts with Performance Drops.** We require that there is substantial performance degradation between the training and test splits, i.e., we observe large drops in performance between the in-distribution and out-of-distribution times.

- **Gradual Temporal Distribution Shifts.** Sudden shifts are well-represented by existing benchmarks on domain shift and subpopulation shift. Models can more effectively extrapolate temporal correlations when the distribution shifts occur gradually over time, as opposed to sudden shifts. Thus, in this paper, we focus on gradual temporal distribution shifts, where we require gradual performance drops between consecutive periods of time.

### 2.2   Evaluation Strategies

Before presenting the datasets, we first discuss two evaluation strategies in Wild-Time.

**Evaluation with a fixed time split (Eval-Fix).** Eval-Fix evaluates models on a single, fixed train-test time split and offers a simple and quick evaluation protocol. Eval-Fix is the primary evaluation strategy in Wild-Time. Concretely, we denote the split timestamp as $t_s$. The train and test sets are $\mathcal{T}^{tr} = \{t \leq t_s | \forall t\}$, $\mathcal{T}^{ts} = \{t > t_s | \forall t\}$, respectively. Eval-Fix evaluates performance using two metrics, average and worst-time performance (Avg) and worst-time performance (Worst). Specifically, let $R_t$ denote the performance at each timestamp $t$. We define the average performance (Avg) and worst-time performance (Worst) as

$$\text{Avg} = \frac{1}{|\mathcal{T}^{ts}|} \sum_{t \in \mathcal{T}^{ts}} R_t, \ \text{Worst} = \min_{t \in \mathcal{T}^{ts}} R_t. \tag{1}$$

Here, average performance measures the overall out-of-distribution performance. Worst-time performance evaluates the model's robustness over time.

**Evaluation with data stream (Eval-Stream).** Eval-Stream evaluates models at each timestamp, evaluating average and worst-time performance on the next $K$ timestamps. Eval-Stream mimics standard machine learning development pipelines, where models are updated frequently and evaluated on timestamps in the near future.

Specifically, we construct a performance matrix $\mathcal{R} \in \mathbb{R}^{T \times K}$, where each element $R_{i,j}$ is the test accuracy of the model trained on timestamp $t_i$ and evaluated on timestamp $t_j$. Following [69], we define the average performance ($\text{Avg}_{\text{stream}}$) and worst-time performance ($\text{Worst}_{\text{stream}}$) as

$$\text{Avg}_{\text{stream}} = \frac{1}{|\mathcal{T}|K} \sum_{t \in \mathcal{T}} \sum_{j=t+1}^{t+K} R_{i,j}, \ \text{Worst}_{\text{stream}} = \frac{1}{|\mathcal{T}|} \sum_{t \in \mathcal{T}} \min_{j \in \{t+1 \ldots t+K\}} R_{i,j} \tag{2}$$

Here, $K$ is a hyperparameter. Compared with typical continual learning metrics in Lopez-Paz and Ranzato [69], we evaluate performance on the next few timestamps, rather than just the subsequent timestamp, to assess the model's robustness across time.

## 3 Datasets

In this section, we briefly discuss the datasets and tasks included in Wild-Time, which reflect natural gradual temporal distribution shifts. We provide more detailed descriptions of all datasets in Appendix A. Additionally, in Appendix F, we discuss some datasets that violate our criteria of dataset selection discussed in Section 2.1, e.g., datasets with sudden temporal distribution shifts.

**Yearbook (Appendix A.1).** Social norms, fashion styles, and population demographics change over time. This is captured in the Yearbook dataset, which consists of 37,921 frontal-facing American high school yearbook photos [26]. We exclude portraits from $1905 - 1929$ due to the limited number of examples in these years, resulting in 33,431 examples from $1930 - 2013$. Each photo is a $32 \times 32 \times 1$ grey-scale image associated with a binary label $y$, which represents the student's gender. In Eval-Fix, the training set consists of data from before 1970, and the test set comprises data after 1970, which corresponds to 40 and 30 years, respectively.

**FMoW-Time (Appendix A.2).** Machine learning models can be used to analyze satellite imagery and aid humanitarian and policy efforts by monitoring croplands [44] and predicting crop yield [93] and poverty levels [42]. Due to human activity, satellite imagery changes over time, requiring models that are robust to temporal distribution shifts.

We study this problem on the Functional Map of the World (FMoW) dataset [16], adapted from the WILDS benchmark [55] and is named as FMoW-Time. Given a satellite image, the task is to predict the type of land usage. The FMoW-Time dataset [55] consists of 141,696 examples from $2002 - 2017$. Each input $x$ is a $224 \times 224$ RGB satellite image, and the corresponding label $y$ is one of 62 land use categories. We use the train/val/test splits in WILDS to construct FMoW-Time dataset in Wild-Time. The train/val/test data splits from WILDS contain images from disjoint location coordinates, and all splits contain data from all 5 geographic regions. In Eval-Fix, the training set includes data from $2002 - 2015$, and the test set includes data from $2016 - 2017$.

**MIMIC-IV (Appendix A.3).** Many machine learning healthcare applications have emerged in the last decade, such as predicting disease risk [70], medication changes [105], patient subtyping [5], in-hospital mortality [30], and length of hospital stay [20]. However, changes in healthcare over time, such as the emergence of new treatments and changes in patient demographics, are an obstacle in deploying machine learning-based clinical decision support systems [30].

We study this problem on MIMIC-IV, one of the largest public healthcare datasets that comprises abundant medical records of over 40,000 patients. In MIMIC-IV, we treat each admission as one record, resulting in 216,487 healthcare records from $2008 - 2019$. To protect patient privacy, the reported admission year is in a three year long date range. Hence, our timestamps are groups of three years: $2008 - 2010, 2011 - 2013, 2014 - 2016, 2017 - 2019$. We consider two classification tasks:

- **MIMIC-Readmission** aims to predict the risk of being readmitted to the hospital within 15 days.
- **MIMIC-Mortality** aims to predict in-hospital mortality for each patient.

For each record, we concatenate the corresponding ICD9 codes [80] of diagnosis and treatment. A binary indicator is used to indicate the codes are come from diagnosis or treatment. We use the concatenated one as the input feature. The label is a binary value that indicates whether the patient is readmitted or passed away for MIMIC-Readmission and MIMIC-Mortality, respectively. For the Eval-Fix setting, the train set consists of patient data fom $2008 - 2013$, while the test set consists of data from $2014 - 2020$.

**Huffpost (Appendix A.4).** In many language models which deal with information correlated with time, temporal distribution shifts cause performance degradation in downstream tasks such as Twitter hashtag classification [45] or question-answering systems [66]. Performance drops across time reflect changes in the style or content of current events.

We study this temporal shift on the Huffpost dataset [76]. The task is to identify tags of news articles from their headlines. Each input feature $x$ is a news headline, and the output $y$ is the news category.

We only keep categories that appear in all years from $2012 - 2022$, resulting 11 categories in total. We choose year 2015 as the split timestamp in Eval-Fix.

**arXiv (Appendix A.5).** Due to the evolution of research fields, the style of arXiv pre-prints also changes over time, reflected by the change in article categories. For example, "neural network attack" was originally a popular keyword in the security community, but gradually became more prevalent in the machine learning community. We study this temporal shift in the arXiv dataset [18], where the task is to predict the primary category of arXiv pre-prints given the paper title as input. The entire dataset includes 172 pre-print categories from $2007 - 2022$.

## 4 Baselines for Temporal Distribution Shifts

Many algorithms have been proposed to improve a model's robustness to distribution shifts or improve a model's performance on a stream of data. For our evaluation, we choose several representative methods from five main categories: classical supervised learning (empirical risk minimization), continual learning, invariant learning, self-supervised learning, and ensemble learning. These methods have been successful on domain generalization and continual learning benchmarks. We extend the selected invariant learning approaches to the temporal distribution shift setting. See Appendix B for a detailed discussion of these algorithms and how we apply them to our tasks.

**Classical Supervised Learning (Appendix B.1).** We evaluate the performance of empirical risk minimization (ERM) on all tasks. In Eval-Fix, we directly train a machine learning model with ERM. In Eval-Stream, we apply ERM to every timestamp and report the performance.

**Continual Learning (Appendix B.2).** Continual learning, also known as lifelong learning or incremental learning, aims to effectively learn from non-stationary distributions via a stream of data [1, 13, 54, 92, 111, 14, 69, 84, 95]. The goal is to accumulate and reuse knowledge in future learning without forgetting information needed for previous tasks, a phenomenon known as catastrophic forgetting [54], which may enable such models to robustly extrapolate into the future in the temporal shift setting. We evaluate four representative algorithms, including fine-tuning, regularization-based (EWC, SI) and memory-based (A-GEM) methods. These methods have been successful on several continual learning benchmarks, such as permuted MNIST [28] and Split TinyImagenet [61].

**Temporally Invariant Learning (Appendix B.3).** Invariant learning methods learn representations or predictors that are invariant across different domains. Common approaches include aligning feature representation over different domains [24, 68, 96, 99, 104, 110, 115], learning invariant predictors via selective augmentation [108] or by strengthening the correlations between representations and labels [2, 3, 53, 58], and optimizing worst-group performance [88, 113, 114]. In Wild-Time, we select four representative invariant learning methods: CORAL [96], IRM [3], LISA [108], and GroupDRO [88]. We adapt these methods to a temporal setting and train them incrementally at each timestamp $t$. In the following, we discuss how we adapt these approaches to the temporal shift setting.

As mentioned in Section 2, in the temporal shift setting, the "timestamp" information is not the same as the notion of a "domain," as the distribution shift may not occur between consecutive timestamps. The data streams in our benchmark are unsegmented and do not include domain boundaries. This setting poses new challenges to the above invariant learning approaches, which rely on domain labels.

To address this challenge, we adapt the above invariant learning approaches to the temporal distribution shift setting. We leverage timestamp metadata to create a temporal robustness set consisting of substreams of data, where each substream is treated as one domain. Specifically, as shown in Figure 3, we define a sliding window $\mathcal{G}$ with length $L$. For a data stream with $T$ timestamps, we apply the sliding window $\mathcal{G}$ to obtain $T - L + 1$ substreams. We treat each substream as a "domain" and apply the above invariant algorithms on the robustness set. We name the adapted CORAL, GroupDRO and IRM as CORAL-T, GroupDRO-T, IRM-T, respectively. Note that we do not adapt LISA since the intra-label LISA performs well without domain information, which is also mentioned in the original paper.

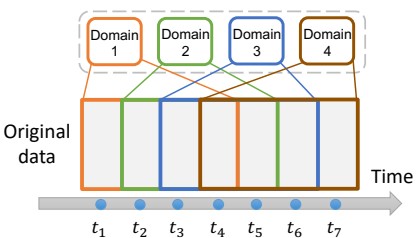

Figure 3: Construction of a temporal robustness set. Here, we have $T = 7$ and $L = 3$. By applying sliding window-based segmentation, we obtain 4 substreams of data. We apply invariant learning approaches to this robustness set.

See Appendix E.2 for details on the temporal adaptation of these invariant learning algorithms and additional ablations. In particular, we compare vanilla invariant learning algorithms with their temporally adapted versions, assess the effect of the length of the time window $L$, and compare performance using overlapping versus non-overlapping time windows.

**Self-Supervised Learning (Appendix B.4).** Self-supervised learning has been shown to improve out-of-distribution robustness [43, 94]. Here, we introduce two representative self-supervised learning methods, SimCLR [15] and SwaV [11], and evaluate their performance on the Wild-Time image classification datasets, Yearbook and FMoW-Time.

**Ensemble Learning (Appendix B.5).** Typically, using ensemble learning improves the performance of machine learning models. We introduce Stochastic Weighted Averaging (SWA) as the representative ensemble learning approach, which is an approximate Bayesian method which averages multiple parameter values along the trajectory of stochastic gradient descent [41].

# 5 Experiments

We benchmark the performance of all methods in Section 4 on each dataset in Wild-Time. Each baseline is evaluated using both the Eval-Fix and Eval-Stream settings. See Appendix D for all evaluation results and experimental details under the Eval-Stream setting.

## 5.1 Experimental Setup

**Data Split.** For both Eval-Fix and Eval-Stream, the training and test sets are subsets of the entire dataset such that the training timestamps are earlier than the test timestamps. We measure temporal out-of-distribution (OOD) robustness as performance on the test set. To compare out-of-distribution with in-distribution (ID) performance, we measure the average per-timestep performance on a held-out set of 10% training examples (20% for MIMIC-Mortality and MIMIC-Readmission) from each training ID timestamp. More details of our data split protocols are described in Appendix C.1.

**Evaluation Metrics.** We measure accuracy in most classification tasks, including Yearbook, FMoW-Time, MIMIC-Readmission, HuffPost, and arXiv. For the MIMIC-Mortality task, we use ROC-AUC due to label imbalance.

**Hyperparameter Settings.** For each dataset, we use the same backbone for all baselines. The choice of backbones are based on the original paper (e.g., DenseNet101 [36] for FMoW-Time [55], or the commonly used ones (e.g., DistilBERT [90] for arXiv and Huffpost). For each method, we tune hyperparameters using cross-validation with grid search. In Eval-Fix, we hold out 10% of the data of each training timestamp (20% for MIMIC-Readmission, and MIMIC-Mortality) to construct the validation set for hyperparameter tuning. Here, we use examples from the remaining 90% of the data to train the model and evaluate the performance on the corresponding validation set. We repeat this process three times via cross-validation with different held-out 10% of the data. After selecting all hyperparameters, we use the entire training set to train the model. See Appendix C.2 for a detailed hyperparameter search setting.

## 5.2 Performance Drops from Temporal Distribution Shifts

The Wild-Time datasets should exhibit observable performance drops between training and test times. In this section, we demonstrate this for every Wild-Time dataset. Table 1 shows the ID and OOD performance of ERM on each Wild-Time dataset. (See Table 20 in the Appendix for comprehensive results on all remaining baselines.)

We observe that OOD performance is substantially lower than ID performance. We conduct further experiments, in which we train ERM on both ID and OOD examples, in Appendix E.1. The substantial drop in ID versus OOD performance indicates that the performance drop is caused by distribution shift rather than the difficulty of training timestamps.

## 5.3 Baseline Comparison

Table 2 shows the performance of all methods on Wild-Time under the Eval-Fix setting. Due to space constraints, we report results on the Eval-Stream setting in Appendix D. We report the average

Table 1: The in-distribution versus out-of-distribution test performance evaluated on Wild-Time under the Eval-Fix setting. For each dataset, higher value means better performance.

| Dataset (Metric) | In-distribution | Out-of-distribution |
|---|---|---|
| Yearbook (Acc) | 97.99 (1.40) | 79.50 (6.23) |
| FMoW-Time (Acc) | 58.07 (0.15) | 54.07 (0.25) |
| MIMIC-Readmission (Acc) | 73.00 (2.94) | 61.33 (3.45) |
| MIMIC-Mortality (AUC) | 90.89 (0.59) | 72.89 (8.96) |
| Huffpost (Acc) | 79.40 (0.05) | 70.42 (1.15) |
| arXiv (Acc) | 53.78 (0.16) | 45.94 (0.97) |

Table 2: The out-of-distribution test performance of each method evaluated on Wild-Time under the Eval-Fix setting. Different groups of rows correspond to different categories of methods. Full table with standard deviation are computed over three random seeds and reported in Table 20 of Appendix. We bold the best OOD performance for each dataset.

| | Yearbook (Accuracy (%) ↑) | | FMoW-Time (Accuracy (%) ↑) | | MIMIC-Readmission (Accuracy (%) ↑) | |
|---|---|---|---|---|---|---|
| | Avg. | Worst | Avg. | Worst | Avg. | Worst |
| Fine-tuning | 81.98 | **69.62** | 44.25 | 37.14 | 62.19 | 59.57 |
| EWC | 80.07 | 66.61 | 44.02 | 36.42 | **66.40** | **64.69** |
| SI | 78.70 | 65.18 | 44.25 | 37.14 | 62.60 | 61.13 |
| A-GEM | 81.04 | 67.07 | 44.10 | 36.02 | 63.95 | 62.66 |
| ERM | 79.50 | 63.09 | **54.07** | **46.00** | 61.33 | 59.46 |
| GroupDRO-T | 77.06 | 60.96 | 43.87 | 36.60 | 56.12 | 53.12 |
| mixup | 76.72 | 58.70 | 53.67 | 44.57 | 58.82 | 57.30 |
| LISA | 83.65 | 68.53 | 52.33 | 43.30 | 56.90 | 54.01 |
| CORAL-T | 77.53 | 59.34 | 49.43 | 41.23 | 57.31 | 54.69 |
| IRM-T | 80.46 | 64.42 | 45.00 | 37.67 | 56.53 | 52.67 |
| SimCLR | 78.59 | 60.15 | 44.76 | 37.00 | n/a | n/a |
| SwaV | 78.38 | 60.73 | 44.92 | 37.17 | n/a | n/a |
| SWA | **84.25** | 67.90 | 54.06 | **46.01** | 59.10 | 56.54 |

| | MIMIC-Mortality (AUC (%) ↑) | | HuffPost (Accuracy (%) ↑) | | arXiv (Accuracy (%) ↑) | |
|---|---|---|---|---|---|---|
| | Avg. | Worst | Avg. | Worst | Avg. | Worst |
| Fine-tuning | 63.37 | 52.45 | 69.59 | 68.91 | 50.31 | 48.19 |
| EWC | 62.07 | 50.41 | 69.42 | 68.61 | **50.40** | **48.18** |
| SI | 61.76 | 50.19 | 70.46 | 69.05 | 50.21 | 48.07 |
| A-GEM | 61.78 | 50.40 | 70.22 | 69.15 | 50.30 | 48.14 |
| ERM | 72.89 | 65.80 | 70.42 | 68.71 | 45.94 | 44.09 |
| GroupDRO-T | 76.88 | **71.40** | 69.53 | 67.68 | 39.06 | 37.18 |
| mixup | 73.69 | 66.83 | **71.18** | 68.89 | 45.12 | 43.23 |
| LISA | 76.34 | 71.14 | 69.99 | 68.04 | 47.82 | 45.91 |
| CORAL-T | **77.98** | 64.81 | 70.05 | 68.39 | 42.32 | 40.31 |
| IRM-T | 76.16 | 70.64 | 70.21 | 68.71 | 35.75 | 33.91 |
| SWA | 69.53 | 60.83 | 70.98 | **69.52** | 44.36 | 42.54 |

and standard deviation of each method's performance across three different random seeds. For each task, we visualize the OOD performance on every test timestamp in Figure 4, and show best ID performance over all approaches as an upper bound on temporally robust performance. The following high-level observations summarize our findings:

- In FMoW-Time, MIMIC-Readmission, and MIMIC-Mortality, model performance degrades with time (Figures 7h, 4c, 4d), as models exhibit higher OOD accuracy on timestamps closer to that of the training data. Such gradual temporal shifts correspond to our motivation in dataset selection, as discussed in Section 2.1. In Yearbook (Figure 7g), performance fluctuates significantly, with models achieving higher OOD accuracy at later timestamps (e.g., $1991 - 1996$) compared to earlier timestamps (e.g., $1981 - 1986$). In HuffPost and arXiv, models achieve the best performance on the earliest test timestamps. Nevertheless, there is a significant gap between the OOD performance

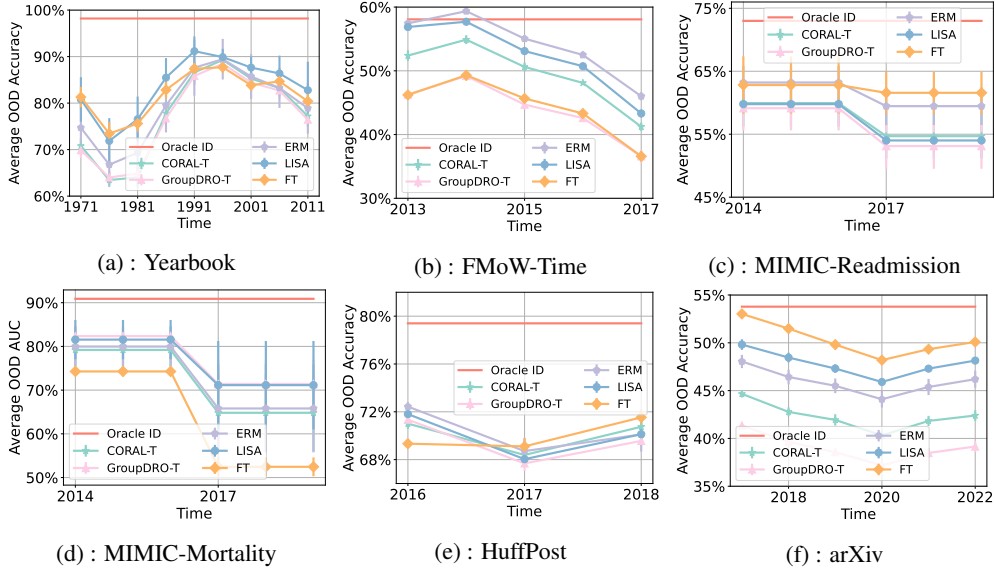

(a) : Yearbook      (b) : FMoW-Time      (c) : MIMIC-Readmission

(d) : MIMIC-Mortality      (e) : HuffPost      (f) : arXiv

Figure 4: Out-of-distribution performance per test timestamp. We select five representative baselines – ERM, FT (Fine-tuning), CORAL-T, GroupDRO-T, LISA, and show the corresponding performance. Oracle ID represent the best ID performance over all compared baselines. Note that for MIMIC-Readmission and MIMIC-Mortality, our OOD timestamps are the three-year blocks $2014 - 2016$, $2017 - 2019$. Hence in Figures 4(c) and 4(d), the performance over this three-year block is the same.

and best ID performance for all datasets and methods. Furthermore, this performance gap changes in a continual manner over time, indicating that the nature of the distribution shift is correlated with the provided timestamps.

- Most invariant learning approaches (CORAL-T, GroupDRO-T, IRM-T, LISA, mixup) did not show clear improvements over ERM. In some cases, invariant learning approaches performed worse than ERM, corroborating the findings in other natural distribution shift benchmarks, such as WILDS [55].

- Incremental training approaches (Fine-tuning, EWC, SI, A-GEM) improve OOD performance on the arXiv and MIMIC-Readmission datasets, and worst OOD performance on the HuffPost dataset. This is expected, since these datasets exhibit more gradual temporal shifts, and incremental training tends to bias the trained model towards the last few timestamps. In all tasks other than Yearbook, incremental training methods perform worse than invariant learning approaches, indicating the power of invariance in learning temporally robust models.

- Neither self-supervised learning nor ensemble learning approaches show consistent benefits over ERM. In summary, ERM has been shown to be a strong baseline in Wild-Time, even when we reduce the number of training examples, as discussed in Appendix E.4.

- Most results from the Eval-Stream setting (Appendix D) concur with the above findings from the Eval-Fix setting. In particular, invariant learning approaches outperform continual learning approaches in more scenarios except FMoW-Time, Huffpost and arXiv, though we do not restrict the buffer size for invariant learning approaches. We hope that Wild-Time will be used to investigate more memory-efficient invariant learning approaches.

## 6   Comparison with Existing Benchmarks

Wild-Time offers a unified framework to facilitate the development of models robust to in-the-wild temporal distribution shifts. We discuss how Wild-Time is related to existing distribution shift and continual learning benchmarks.

**Relation to Distribution Shift Benchmarks.** Distribution shift has been widely studied in the machine learning community. Early works presented small-scale benchmarks to study distribution shifts in sentiment analysis [22] and object detection [87]. Subsequent distribution shift benchmarks focused on larger-scale, real-world data. The first line of such benchmarks induce distribution shifts

by applying different transformations to object recognition datasets. These benchmarks include: (1) ImageNet-A [35], ImageNet-C [33], and CIFAR-10.1 [85], which add noise or adversarial examples to the original Imagenet [86] and CIFAR [57] datasets, respectively; (2) Colored MNIST [3], which changes the color of digits from the original MNIST. More recent works created domain generalization benchmarks by collecting sets of images with different styles or backgrounds, such as PACS [62], DomainNet [82], VLCS [21], OfficeHome [102], ImageNet-R [34], BREEDS [91], Waterbirds [88], NICO [32], and MetaShift [65]. While these datasets are useful testbeds for verifying the efficacy of new algorithms, they do not reflect natural distribution shifts that arise in real-world applications.

Recently, a few works have constructed datasets and benchmarks for real-world distribution shifts. WILDS benchmark consists of ten datasets spanning a wide range of real-world applications, such medical image recognition, sentiment classification, land-use classification with satellite image, and code autocompletion, with a focus on domain shifts and subpopulation shifts [55]. WILDS 2.0 extends WILDS and introduces unlabeled data to help boost model robustness to distribution shifts [89]. SHIFTS [72] is composed of three datasets, concerning weather prediction, machine translation, and self-driving vehicle motion prediction. Unlike these works that focus on general distribution shifts, we target temporal distribution shifts arising in real-world applications. A few recent works have started investigating model robustness over time, in real-world applications such as healthcare-related prediction [30], drug discovery [37], image-based geo-localization [10], machine reading comprehension [66], and tweet hashtag prediction [45]. Unlike prior datasets that target specific applications, Wild-Time presents a comprehensive benchmark comprised of 7 datasets from diverse domains and offers systematic evaluation protocols.

**Relation to Continual Learning Benchmarks.** Continual learning methods are often benchmarked on image classification datasets. Some popular benchmarks such as RainbowMNIST [23] and permuted MNIST [50] apply various image transformations to a small-scale image dataset to obtain a sequence of tasks. Others such as Split CIFAR100 [57], Split TinyImagenet [61], F-CelebA [51], and Stanford Cars [56] split a large image dataset into multiple non-overlapping class sets, where each is regarded as one task. A third collection of related benchmarks treats each object recognition dataset as a different task. For example, Visual Domain Decathlon [64] consists of 10 datasets from various domains, such as Aircraft [71], SVHN [77], Omniglot [59], VGG-Flowers [78], CLEAR [67]. In the natural language processing (NLP) domain, continual learning benchmarks such as ASC [52] and DSC [51] have been used to evaluate the performance of large-scale pretrained models over time. Unlike these prior benchmarks, Wild-Time presents a collection of datasets that reflect natural temporal distribution shifts arising in real-world applications as well as an evaluation strategy (Eval-Stream) to assess incremental learning approaches.

## 7 Conclusion and Discussion

In this paper, we present the Wild-Time benchmark, and examine in-the-wild distribution shifts over time. We leverage timestamp metadata, which is largely ignored by existing robustness techniques and benchmarks. Wild-Time includes 6 tasks from 5 datasets, which span a range of applications (facial recognition, news, healthcare) and tasks (classification, regresssion). On each of these datasets, we systematically benchmark 13 approaches, including continual learning, invariant learning, self-supervised learning, and ensemble learning approaches. Our experiments show a large gap between ID and OOD performance on all tasks due to temporal distribution shift. We conclude that no existing invariant learning, continual learning, self-supervised, or ensemble learning approach is consistently more robust to temporal distribution shifts than ERM. We hope that Wild-Time facilitates further research in developing temporally robust methods that can be safely deployed in the wild.

## Acknowledgement

We thank Zhenbang Wu, Shiori Sagawa, Kexin Huang, Ananya Kumar, Scott Lanyon Fleming, and members of the IRIS lab for the many insightful discussions and helpful feedback. This research was funded in part by Apple, Intel, Juniper Networks, and JPMorgan Chase & Co. Any views or opinions expressed herein are solely those of the authors listed, and may differ from the views and opinions expressed by JPMorgan Chase & Co. or its affiliates. This material is not a product of the Research Department of J.P. Morgan Securities LLC. This material should not be construed as an individual recommendation for any particular client and is not intended as a recommendation of particular securities, financial instruments or strategies for a particular client. This material does not constitute a solicitation or offer in any jurisdiction. CF is a CIFAR fellow.

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
