# OpenReview forum: "Wild-Time: A Benchmark of in-the-Wild Distribution Shift over Time"
_NeurIPS.cc/2022/Track/Datasets_and_Benchmarks — NeurIPS 2022 Datasets and Benchmarks _

### Official Review · Reviewer_rLDr · 2022-07-23
**A paper with good contributions to the distributional shift problem**

**Rating:** 8
**Confidence:** 4

**Strengths:**

* While most previous distributional shift benchmarks artificial distributional shifts (e.g., image corruptions [21], background [63], color [3]), the proposed Wile-Time benchmark focuses on the natural temporal distributional shift, which is a problem that naturally exists in the real-world. This is a significant contribution to this community and can facilitate future works to develop robust algorithms against temporal distribution shift.

* The new benchmark is comprehensive in terms of evaluation metrics, evaluation methods, and diversity of datasets.

**Weaknesses:**

It will be better to discuss potential research directions to address this problem.

**Additional Feedback:**

No additional feedback.

**Clarity:**

Yes. The paper is well-written.



Typo: L248: “domain”. -> “domain.” (The period should be inside of the quote; the right quotation mark is wrong)

**Correctness:**

In general, the dataset is constructed in a sound way and evaluations are designed properly.

However, I am confused by the adaptation of the invariant learning methods (L241-254). If I understand correctly, data from each time window will be regarded as one domain. However, as shown in Fig. 3, the time windows are overlapped with each other. Therefore, does it mean that one data point can belong to multiple domains? This is quite different from the traditional definition of domain, e.g., on Waterbirds [63], an image of waterbird on water background is regarded as one domain in GroupDRO [63] and does not belong to other domains. Can authors clarify why extending the invariant learning methods this way and why not using non-overlapping time windows?

**Documentation:**

Yes, the dataset and code are released.

**Ethics:**

No ethical concerns.

**Relation To Prior Work:**

Yes. The authors discuss the difference between Wild-Time and two types of benchmarks—distributional shift benchmarks and continual learning benchmarks.

**Summary And Contributions:**

The paper proposes Wild-Time benchmark of seven datasets to benchmark methods under temporal distributional shift. Two evaluation settings are proposed. The first evaluation setting is Eval-Fix, which uses a fixed training and testing split as in supervised learning, which is usable to broader research communities. The second evaluation setting is Eval-Stream, which is closer to the continual learning evaluation setting. The paper benchmarks two categories of approaches, domain generalization methods and continue learning methods. The results show that existing methods cannot address the temporal distributional shift problem well, leaving room for future methods to address this problem.

---

> ### Author Response · Authors · 2022-08-22
> **Response to Reviewer rLDr**
>
> Thank you for your insightful and constructive feedback. We address all of your comments in our response below. Please let us know if you have any additional questions or concerns.
>
> **Q1**: However, I am confused by the adaptation of the invariant learning methods (L241-254). If I understand correctly, data from each time window will be regarded as one domain. However, as shown in Fig. 3, the time windows are overlapped with each other. Therefore, does it mean that one data point can belong to multiple domains?This is quite different from the traditional definition of domain, e.g., on Waterbirds [63], an image of waterbird on water background is regarded as one domain in GroupDRO [63] and does not belong to other domains. Can authors clarify why extending the invariant learning methods this way and why not using non-overlapping time windows?
>
> **A1**: We chose to allow data points to belong to multiple domains in order to learn the gradual temporal distribution shift in the Wild-Time datasets. However, we trained all invariant learning baselines on non-overlapping windows, and report the OOD average performance in Table Re-1. See the full table in Table 16 of Appendix and the corresponding descriptions in Appendix E.3.3 in the revised version.
>
> **Table Re-1**: Performance of CORAL-T, GroupDRO-T, and IRM-T baselines when trained on overlapping and non-overlapping time substreams.
>
> |            |                 |    Yearbook   |      FMoW     | MIMIC-Readmission | MIMIC-Mortality |
> |------------|-----------------|:-------------:|:-------------:|:-----------------:|:---------------:|
> | CORAL-T    | Overlapping     |  **77.53 (2.15)** |  **47.34 (0.09)** |    **64.50 (3.03)**   |   **70.81 (3.22)**  |
> |            | Non-overlapping |  75.97 (0.63) |  47.28 (0.74) |    57.42 (9.44)   |   61.21 (0.45)  |
> | GroupDRO-T | Overlapping     |  **77.06 (1.67)** |  **37.61 (1.16)** |   **66.91 (0.91)**   |   **73.06 (2.32)**  |
> |            | Non-overlapping |  76.94 (1.87) |  35.05 (0.25) |    47.92 (8.71)   |   61.42 (0.21)  |
> | IRM-T      | Overlapping     |  **80.46 (3.53)** |  38.73 (1.67) |    **59.67 (2.19)**   |   **67.02 (4.37)**  |
> |            | Non-overlapping |  77.21 (2.34) |  **45.89 (0.21)** |    48.04 (9.35)   |   61.10 (0.14)  |
> |            |                 |    **Drug-BA**    | **Precipitation** |      **Huffpost**     |      **arXiv**      |
> | CORAL-T    | Overlapping     | **0.355 (0.016)** |  **46.97 (0.60)** |    **70.05 (0.63)**   |   **42.32 (0.60)**  |
> |            | Non-overlapping | 0.351 (0.008) |  45.36 (1.93) |    68.11 (1.40)   |   42.07 (0.72)  |
> | GroupDRO-T | Overlapping     | 0.342 (0.006) |  **45.89 (0.44)** |    **69.53 (0.54)**   |   **39.06 (0.54)**  |
> |            | Non-overlapping |  **0.361 (0.08)** |  45.75 (0.72) |    68.41 (0.41)   |   36.07 (1.35)  |
> | IRM-T      | Overlapping     |  0.355 (0.04) |  46.38 (1.61) |    **70.21 (1.05)**   |   35.75 (0.90)  |
> |            | Non-overlapping | **0.362 (0.005)** |  **46.54 (0.86)** |    69.58 (0.79)   |   **38.85 (0.44)**  |
>
>
> For the Yearbook, Huffpost, arXiv, MIMIC-Mortality-WildT, MIMIC-Readmission-WildT, and Precipitation-WildT datasets, invariant learning baselines generally obtained better performance using overlapping time windows. For Drug-BA, we obtained better performance using non-overlapping time windows. This is what we expected since overlapping window indeed maintain the correlations between different timesteps, which maybe more suitable to model gradual temporal distribution shift. Since, on the aggregate, using overlapping time windows resulted in better performance, we keep the results using non-overlapping windows in Table 2 of the main paper.
>
> **Q2**: Typo: L248: “domain”. -> “domain.” (The period should be inside of the quote; the right quotation mark is wrong)
>
> **A2**: Thank you for pointing it out. We’ve corrected the typo.

---

> > ### Comment · Reviewer_rLDr · 2022-08-28
> > **Question well answered**
> >
> > The authors' response addresses my confusion about overlapping vs. non-overlapping windows. The ablation study in Tab. 16 and Appendix E.3.3 demonstrate the advantage of using overlapping windows when considering the results on multiple datasets.
> >
> > Overall, the paper makes a good contribution to the OOD community. I raised my rating to "8: Top 50% of accepted papers, clear accept."

---

> > > ### Author Response · Authors · 2022-08-28
> > > **Response to Reviewer rLDr**
> > >
> > > Dear Reviewer rLDr,
> > >
> > > Thank you for your response and for raising your rating, we are happy to see that our response addresses your concerns.

---

### Official Review · Reviewer_crkT · 2022-07-24

**Rating:** 7
**Confidence:** 4

**Strengths:**


- So far, the domain of continual learning has been mainly used to employ traditional datasets such as CIFAR and ImageNet datasets by gradually increasing class information and manipulating available datapoints. Many researchers have been encouraged by this point to easily pursue continual learning studies. However, this setting is somewhat far from real-world circumstances. Accordingly the authors provide datasets that matches the real-world problems.

- The proposed dataset was evaluated by several baselines recently utilized in distribution shift and continuous learning to confirm the properties by comparing their performance, allowing researchers who need to use the data to do so effectively.

- Through the baselines and extensive experiments, the proposed datasets clearly follow characteristics of both continual learning and distributional shifts.


**Weaknesses:**

- This work tells that invariant learning algorithms have a high level of performance. I guess that the hyper-parameters; K and T, were well calibrated. If the appendix results are provided in the manuscript, there appears to be no meaningful difference as compared to continuous learning. I'm wondering how invariant learning algorithms properly work when decreasing values of K and T. In this regard, I have a doubt whether the invariant learning algorithms observes more data-points in the same batch or epoch comparing the cases of continual learning baselines.

- I might think that the provided data adequately describe the distribution shift. However, when I look at the table 1 and table 12, it seems that the "Empirical Risk Minimization" work well, allowing the out of distribution dataset to be covered with in-distribution data. When the number of available datapoints is reduced, it would be possible to demonstrate that ERM is ineffective. If it could be demonstrated that the existing distributional shift and continuous learning methods perform better than ERM on smaller datasets, then ERM would be rendered obsolete. I believed that it was originally intended.

**Additional Feedback:**

I'm awaiting the outcomes of self-supervised approaches and ablation studies based on the reduction of available datapoints. I'm able to to increase my score to 7 based on author's response to my comment.


---

**After rebuttal :** After interacting with the authors and considering the comments and revisions, I am happy to raise my score. The paper can make a good contribution to this venue.

**Clarity:**

The writing and presentation are quite adequate, with explanation of each dataset in the appendix.
However, it takes multiple reads to get a sense of the contributions, and the figures and details are not provided to facilitate quick comprehension of datasets. Specifically, they must demonstrate graphically the reason why the proposed datasets such as drug discovery, climate dataset and health care dataset follow distributional shifts like yearbook and FMoW. Although this is not my major factor in my evaluation of this paper, I believe that authors should dedicate to improve the paper readability.

**Correctness:**

- While the baselines in this paper were picked in consideration of various domains, self-supervised learning approaches that recently get high attention due to the best performance are not considered. Despite the fact that these methods are limited to use for image classification tasks, at least, the author had to apply those to the "yearbook" and "FMow" datasets considering the present studies.

- The Eval Fix and Eval-Stream evaluation metrics are well-designed. It is because the majority of continuous learning has shown no interest in time-variants, and distributional shift does not place a priority on out-of-sample data due to the time-effect.

**Documentation:**

The appendix adequately describes how all datasets are constructed and includes instructions for manipulating all datasets' base algorithms.
However, when I observe the source codes on GitHub, the authors require to make an effort to explain how to run these codes and assess baselines. In particular, they should provide the pre-trained checkpoints used in the manuscript and appendix, as additional experimental results demonstrate that this datasets reduce gaps between synthetic datasets and real-world applications.

**Ethics:**

There are no ethical concerns about this work from my side.

**Relation To Prior Work:**

The previous dataset, Wild, focused on image classification tasks, meanwhile the proposed dataset, Wild-Time includes multiple domains including drug discovery, news classification (arxiv dataset) and climate issues; precipitation.

**Summary And Contributions:**


- This paper proposes a dataset for distribution shifts that incorporates the passage of time using timestamp metadata.
This reflects natural temporal distribution shifts. Consequently, the proposed dataset, Wild-Time, can be viewed as a more plausible dataset for real-world situations.
- While the previous dataset (Wild) concentrated on image datasets, this paper deals with various datasets including drug discovery, health care and news classification.
- Moreover, they suggest assessment of the proposed dataset based on the manipulation of timestamp metadata known as Eval-Fix; time-series setting  and Eval-Stream; continual learning setting.
- To demonstrate the efficacy of the proposed dataset, the authors conducted extensive experiments. They consistently tell us that the proposed dataset represents distinct properties of both in-distribution and out of distribution dataset.

---

> ### Author Response · Authors · 2022-08-22
> **Response to Reviewer crkT (1/4)**
>
> Thank you for your constructive and insightful feedback. We added the additional baselines and ablation studies in response to your comments and discuss these changes below. Please let us know if our response addresses all of your concerns.
>
> **Q1**: This work tells that invariant learning algorithms have a high level of performance. I guess that the hyper-parameters; K and T, were well calibrated. If the appendix results are provided in the manuscript, there appears to be no meaningful difference as compared to continuous learning. I'm wondering how invariant learning algorithms properly work when decreasing values of K and T.
>
> **A1**: In our paper, T refers to the total number of timestamps in the dataset, and is fixed. By $K$, do you mean $L$, the size of the time window (defined in Lines 265-267 of the revised paper)?
>
> We included an ablation in which we report CORAL-T, GroupDRO-T, and IRM-T performance when $L$ is reduced and report the results in Table Re-1. We also add these new results in Section 3.2 of Appendix in the revised paper. According to the results, we find that reducing L marginally worsens performance of invariant learning baselines.
>
> Table Re-1: Performance of the temporally adapted invariant learning baselines when decreasing the length of the time windows, $L$ (See full table in Appendix E.3.2).
>
> |            |   |    Yearbook   |     |     FMoW     |
> |------------|:-:|:-------------:|:---:|:------------:|
> |            | L |    OOD Avg    |  L  |    OOD Avg   |
> | GroupDRO-T | 5 |  **77.06 (1.67)** |  3  | **37.61 (1.16)** |
> |            | 4 |  72.84 (3.04) |  2  | 30.67 (0.46) |
> |            | 2 |  73.42 (2.29) | n/a |      n/a     |
> | CORAL-T    | 5 |  **77.53 (2.15)** |  3  | **47.34 (0.09)** |
> |            | 4 |  77.09 (1.56) |  2  | 47.05 (0.20) |
> |            | 3 |  76.92 (1.07) | n/a |      n/a     |
> | IRM-T      | 5 |  **80.46 (3.53)** |  3  | 38.73 (1.67) |
> |            | 4 |  79.56 (3.12) |  2  | **39.22 (0.33)** |
> |            | 2 |  79.47 (2.69) | n/a |      n/a     |
> |            |   | **Precipitation** |     |     **arXiv**    |
> |            | L |    OOD Avg    |  L  |    OOD Avg   |
> | GroupDRO-T | 4 |  **45.89 (0.44)** |  4  | **39.06 (0.54)** |
> |            | 2 |  44.59 (0.49) |  2  | 38.15 (0.77) |
> | CORAL-T    | 4 |  **46.97 (0.60)** |  4  | **42.32 (0.60)** |
> |            | 2 |  45.98 (0.61) |  2  | 41.59 (0.58) |
> | IRM-T      | 4 |  **46.38 (1.61)** |  4  | **35.75 (0.90)** |
> |            | 2 |  46.23 (1.33) |  2  | 31.76 (0.49) |
>
> $K$ denotes the number of future timestamps we evaluate on in the Eval-Stream setting. If you mean $K$, please let us know and we are happy to conduct additional experiments.

---

> > ### Author Response · Authors · 2022-08-22
> > **Response to Reviewer crkT (2/4)**
> >
> > **Q2**: I might think that the provided data adequately describe the distribution shift. However, when I look at the table 1 and table 12, it seems that the "Empirical Risk Minimization" work well, allowing the out of distribution dataset to be covered with in-distribution data. When the number of available datapoints is reduced, it would be possible to demonstrate that ERM is ineffective. If it could be demonstrated that the existing distributional shift and continuous learning methods perform better than ERM on smaller datasets, then ERM would be rendered obsolete. I believed that it was originally intended.
> >
> > **A2**: ERM has been shown to be a strong baseline, especially under domain shift, as discussed in [Koh et al. ICML 2021]. We add an ablation by reducing the number of training examples at each ID timestamp and report results for all baselines. Specifically, we randomly allocate 30% of the data at each timestamp as training, rather than 90% in our original results. The results with averaged OOD performance are reported in Table Re-2 and we put the full table with detailed discussion in Appendix E.5.
> >
> > **Table Re-2**: Performance (OOD Avg.) of all baselines when reducing the amount of training data (see Appendix E.5 for full results and detailed discussion).
> >
> > |             | Yearbook |      FMoW     | MIMIC-Readmission | MIMIC-Mortality |
> > |-------------|:--------:|:-------------:|:-----------------:|:---------------:|
> > | Fine-tuning |   52.00  |     38.93     |       **65.88**       |      58.39      |
> > | EWC         |   48.84  |     39.24     |       45.37       |      59.28      |
> > | SI          |   47.03  |     39.48     |       56.20       |      57.70      |
> > | A-GEM       |   46.98  |     39.17     |       57.51       |      58.63      |
> > | ERM         |   **77.05**  |     46.54     |       40.40       |      60.69      |
> > | GroupDRO-T  |   60.45  |     29.96     |       59.90       |      60.54      |
> > | Mixup       |   77.31  |     **48.22**     |       41.57       |      **61.21**      |
> > | LISA        |   74.17  |     46.60     |       40.48       |      61.09      |
> > | CORAL-T     |   59.66  |     39.48     |       45.00       |      59.75      |
> > | IRM-T       |   60.45  |     38.00     |       52.00       |      60.54      |
> > |             |  **Drug-BA** | **Precipitation** |      **Huffpost**     |      **arXiv**      |
> > | Fine-tuning |   0.239  |     44.96     |       14.12       |      **48.88**      |
> > | EWC         |   0.248  |     45.28     |       13.65       |      48.56      |
> > | SI          |   0.243  |     27.29     |       14.22       |      **48.88**      |
> > | A-GEM       |   0.228  |     34.46     |       **15.53**       |      48.79      |
> > | ERM         |   0.345  |     **47.16**     |       12.32       |      46.07      |
> > | GroupDRO-T  |   **0.358**  |     45.70     |       11.82       |      39.71      |
> > | Mixup       |   0.346  |     46.02     |       13.35       |      45.98      |
> > | LISA        |    n/a   |     45.80     |       13.26       |      47.66      |
> > | CORAL-T     |   0.338  |     46.53     |       13.15       |      42.72      |
> > | IRM-T       |   0.356  |     45.69     |       11.39       |      35.85      |
> >
> > We find that ERM still outperforms invariant learning and continual learning baselines, even with fewer training examples. This corroborates our initial findings, which used more training examples.

---

> > > ### Author Response · Authors · 2022-08-22
> > > **Response to Reviewer crkT (3/4)**
> > >
> > > **Q3**: While the baselines in this paper were picked in consideration of various domains, self-supervised learning approaches that recently get high attention due to the best performance are not considered. Despite the fact that these methods are limited to use for image classification tasks, at least, the author had to apply those to the "yearbook" and "FMow" datasets considering the present studies.
> > >
> > > **A3**: We added two representative self-supervised learning approaches, SwAV [Caron et al. NeurIPS 2020] and SimCLR [Chen et al. ICML 2020], and benchmark them on the image datasets, Yearbook and FMoW. Following [Chen et al. ICML 2020], we first leverage these self-supervised learning approaches to learn good representations, and then use the same training data to fine-tune the classifier. We report the results in Table Re-3 below, where the performance of ERM is also reported. (Full results with all baselines are reported in Table 2 of our revised paper.)
> > >
> > > **Table Re-3**: Performance (accuracy with standard deviation) of self-supervised learning approaches on image classification.
> > >
> > > |        | Yearbook (Accuracy $\uparrow$) |              | FMoW (Accuracy $\uparrow$) |              |
> > > |--------|:------------------------------:|:------------:|:--------------------------:|:------------:|
> > > |        |              Avg.              |     Worst    |            Avg.            |     Worst    |
> > > | ERM    |          **79.50 (6.23)**          | **63.09 (5.15)** |        **51.99 (0.37)**        | **48.79 (0.49)** |
> > > | SimCLR |          78.59 (2.72)          | 60.15 (3.48) |        42.91 (0.40)        | 39.54 (0.67) |
> > > | SwAV   |          78.38 (1.86)          | 60.73 (1.08) |        49.53 (0.27)        | 46.31 (0.58) |
> > >
> > > We observe that these self-supervised learning approaches do not show significant performance gains over ERM, corroborating our main conclusion.
> > >
> > > ---
> > >
> > > **Q4**: The writing and presentation are quite adequate, with explanation of each dataset in the appendix. However, it takes multiple reads to get a sense of the contributions, and the figures and details are not provided to facilitate quick comprehension of datasets. Specifically, they must demonstrate graphically the reason why the proposed datasets such as drug discovery, climate dataset and health care dataset follow distributional shifts like yearbook and FMoW. Although this is not my major factor in my evaluation of this paper, I believe that authors should dedicate to improve the paper readability.
> > >
> > > **A4**: We re-drew Figure 1 to illustrate the types of distribution shifts in both the Yearbook (image classification) and Drug-BA (healthcare/medicine) datasets.
> > >
> > > - We reorganized the Experiments section to improve readability. We replaced Table 1 in the initial version with smaller tables that convey self-contained stories. Specifically, we made the following changes:
> > > - We split ID and OOD performance, and use one subsection (Section 5.2) to discuss the performance drop between ID and OOD datasets with the analysis of task difficulty
> > > - We remove the standard deviations in Table 1 and include the full table in Appendix E.
> > >
> > > We also reorganized the Appendix to make it more readable by reorganizing tables and figures to make them more close to the corresponding descriptions.
> > >
> > > ---
> > >
> > > **Q5**: The appendix adequately describes how all datasets are constructed and includes instructions for manipulating all datasets' base algorithms. However, when I observe the source codes on GitHub, the authors require to make an effort to explain how to run these codes and assess baselines. In particular, they should provide the pre-trained checkpoints used in the manuscript and appendix, as additional experimental results demonstrate that this datasets reduce gaps between synthetic datasets and real-world applications.
> > >
> > > **A5**: We updated the GitHub README with the following changes:
> > > 1. added a script that users can run to download all Wild-Time datasets;
> > > 2. included detailed instructions for running all baselines on the datasets;
> > > 3. stated the licenses under which we are re-sharing the Wild-Time datasets.
> > >
> > > We are preparing model checkpoints and will push them to the Wild-Time Github before the end of the discussion period.

---

> > > > ### Author Response · Authors · 2022-08-22
> > > > **Response to Reviewer crkT (4/4)**
> > > >
> > > > **References**
> > > >
> > > > [Koh et al. ICML 2021] Koh, Pang Wei, Shiori Sagawa, Henrik Marklund, Sang Michael Xie, Marvin Zhang, Akshay Balsubramani, Weihua Hu et al. "Wilds: A benchmark of in-the-wild distribution shifts." In International Conference on Machine Learning, pp. 5637-5664. PMLR, 2021.
> > > >
> > > > [Chen et al. ICML 2020] Chen, Ting, Simon Kornblith, Mohammad Norouzi, and Geoffrey Hinton. "A simple framework for contrastive learning of visual representations." In International conference on machine learning, pp. 1597-1607. PMLR, 2020.
> > > >
> > > > [Caron et al. NeurIPS 2020] Caron, Mathilde, Ishan Misra, Julien Mairal, Priya Goyal, Piotr Bojanowski, and Armand Joulin. "Unsupervised learning of visual features by contrasting cluster assignments." Advances in Neural Information Processing Systems 33 (2020): 9912-9924.

---

> ### Comment · Reviewer_crkT · 2022-08-28
> **Response to authors comment**
>
> I have checked the answers and the revision of manuscript. I feel satisfactory, so that I raise my score to 7.
>
> Lastly, in this revision, authors claim that the current baselines such as continual learning, invariant learning and self-supervised learning, do not have effectively impact on addressing realistic temporal variant problem. During the remaining discussion phase, I hope the authors to reveal which research direction is necessarily proposed to solve in terms of realistic temporal shift.

---

> > ### Author Response · Authors · 2022-08-28
> > **Response to Additional Comments of Reviewer crkT**
> >
> > Thank you for your response and for raising your rating. We’re happy that our response addresses most of your concerns. We reply to your additional comment below:
> >
> > **Q1**: Lastly, in this revision, authors claim that the current baselines such as continual learning, invariant learning and self-supervised learning, do not have effectively impact on addressing realistic temporal variant problem. During the remaining discussion phase, I hope the authors to reveal which research direction is necessarily proposed to solve in terms of realistic temporal shift.
> >
> > **A1**: We believe that there are two aspects that are important to consider in resolving natural distribution shift:
> >
> > - **Learning Changeable Temporal Invariance**. To build a robust model, it would be useful to learn invariance, which captures features in the data that remain invariant across different distributions. However, this is difficult to do when temporal distribution shift happens, as such invariance can also change over time, where one kind of invariance is only suitable for a specific time window. Capturing the correlations between different time windows and determining when and how to update the invariant model are crucial.
> >
> > - **Leveraging supervised and unsupervised adaptation**. In addition to maintaining a temporally invariant model, adapting to new timestamps is also necessary in tackling temporal distribution shifts. Here, we can leverage labeled data from timestamps in the near past and unlabeled observations from the current timestamp to fine-tune the model. How to combine temporal invariance with supervised and unsupervised adaptation to achieve effective adaptation remains an open problem.
> >
> > We’ve added these discussions in Appendix F.3 of the revised version.

---

### Official Review · Reviewer_UtTF · 2022-07-25
**Well addressing an important practical issue in machine learning, however the paper is not practical enough**

**Rating:** 7
**Confidence:** 3

**Strengths:**

Addressing a real-world problem in machine learning is important to be solved for industrial machine learning systems.
The purpose of the paper is clear.
There is no synthetic data creation or changes since the domains are based on actual time frames over time.
The scope of the paper is highly relevant to current machine learning research borders.
The evaluation method is described clearly.
Results are reflected in the paper in a clean and understandable way.
The dataset contains various domains with two different tasks.





**Weaknesses:**

Although the contribution is clear, there is no real-world scenario in the introduction for introducing the importance of the problem to the reader.
All variables in the formula were not defined.
There is no practical sample for how to use this benchmark or any practical guideline for users.
There is no introduction for any framework or library which can implement the benchmark in practical projects.
I couldn't find any link to the dataset or benchmark tool.
The evaluation metrics part is not well described.
There is no ethics section or any consideration about that.





**Additional Feedback:**

-

**Clarity:**

The paper is clear and easy to understand.


**Correctness:**

The evaluation method has been defined completely and tested for evaluating different existing methods.


**Documentation:**

There is no reference to the website or any repo for accessing the dataset or any possible framework to use the dataset and benchmark in practice.

**Ethics:**

There are no ethical concerns in the discussion or any part of the paper.
Ethics should be mentioned in the paper.

**Relation To Prior Work:**

The contribution is clearly mentioned however there is not enough explanation about previous works on time-based shifts.
There is not enough explanation on how temporal shift is completely a new and not addressed field in the data shift area.


**Summary And Contributions:**

This paper proposes a benchmark for addressing the temporal shifts in machine learning systems.
(++) The motivation of the paper is clear. The authors bring up a slight problem of large-scale machine learning systems where the data changes over time.
 (+) The authors bring a practical suggestion for evaluating new models over time-based shifts.
(-) The definition of temporal shifts is very brief, so the importance of these shifts in practice is not well explained.
(--) Since this paper aims to propose a benchmark for mitigating the temporal shifts, It should contain a brief definition of the meaning and effects of this type of shift in each dataset. If there is any difference and more detail, clarify the difference between this type of shift and other data shifts.

---

> ### Author Response · Authors · 2022-08-22
> **Response to Reviewer UtTF**
>
> Thank you for your helpful and valuable feedback. We address your comments below, and discuss the revisions we made accordingly. Please let us know if our response addresses all of your concerns.
>
> **Q1**: Although the contribution is clear, there is no real-world scenario in the introduction for introducing the importance of the problem to the reader.
>
> **A1**: In our initial version, we discussed real-world scenarios in Lines 32-35 about yearbook portraits of American high schoolers. In the new version, we added more real-world scenarios to the introduction to further motivate the problem of temporal robustness, i.e., temporal shifts in real-world ML pipelines (Lines 25-28), temporal shifts in language models (Lines 28-30), flu incidence prediction (Lines 31-32), yearbook portraits of American high schoolers (Lines 36-37 and Figure 1(a)), drug-target interaction prediction (Line 38-40 and Figure 1(b)).
>
> ---
>
> **Q2**: All variables in the formula were not defined.
>
> **A2**: We defined all variables in the formula in Lines 98-102 and Lines 109-111 in our initial version (Lines 103-106 and 110-117 in the revised version).
>
> ---
>
> **Q3**: There is no practical sample for how to use this benchmark or any practical guideline for users. There is no introduction for any framework or library which can implement the benchmark in practical projects. I couldn't find any link to the dataset or benchmark tool.
>
> **A3**: We included a link for downloading all Wild-Time datasets in our initial submission. To make our benchmark more user-friendly, we’ve made the following changes.
> - We included detailed instructions in the Github for downloading all datasets and running baselines.
> - We added a script that users can run to automatically download all Wild-Time datasets.
>
> We are currently working on packaging the Wild-Time benchmark and building APIs so that users can easily load Wild-Time into their projects.
>
> ---
>
> **Q4**: The evaluation metrics part is not well described.
>
> **A4**: We tried to make it more clear by revising some wording in Section 2 (Lines 107-108, 112-113). Please kindly let us know if you have any further concerns.
>
> ---
>
> **Q5**: There is no ethics section or any consideration about that.
>
> **A5**: We already discussed ethical considerations for the Yearbook dataset in Appendix F.2 of our initial submission. We added ethics discussions for the FMoW-WildT and MIMIC-WildT datasets in Appendix F.2 in the revised version.
>
> ---
>
> **Q6**: The contribution is clearly mentioned however there is not enough explanation about previous works on time-based shifts. There is not enough explanation on how temporal shift is completely a new and not addressed field in the data shift area. The definition of temporal shifts is very brief, so the importance of these shifts in practice is not well explained. (--) Since this paper aims to propose a benchmark for mitigating the temporal shifts, It should contain a brief definition of the meaning and effects of this type of shift in each dataset. If there is any difference and more detail, clarify the difference between this type of shift and other data shifts.
>
> **A6**: In our paper, we specifically consider temporal distribution shifts, which has not been systematically studied in prior distribution shift benchmarks, such as WILDS [Koh et al. ICML 2021] and Shifts [Malinin et al. NeurIPS 2021]. We made the following changes in the introduction to provide more examples and explanations on temporal distribution shift:
> - Added two additional real-world examples of temporal distribution shift, i.e., temporal shifts in language models (Lines 28-30), flu incidence prediction (Lines 31-32)
> - Adjusted Figure 1 to convey more examples and visualized the performance drops over time.
> - Highlighted the difference between Wild-Time and other continual learning benchmarks and distribution shift benchmarks in Lines 49-59.
>
> In addition to these changes in the introduction, we already discussed the temporal shifts for every dataset: Yearbook (Lines 123-124), FMoW-WildT (Lines 135-136), MIMIC-IV-WildT (Lines 150-152), Drug-BA (Lines 169-170), Precipitation-WildT (Lines 177-179), Huffpost (Lines 186-187), arXiv (Lines 192-195).
>
> We find that both continual learning and time-invariant learning approaches cannot effectively tackle natural temporal shifts. As temporal shift is still an underrepresented field in distribution shifts, we believe that the proposed Wild-Time benchmark can help facilitate future research on temporal shifts.
>
> ---
>
> **References**
>
> [Koh et al. ICML 2021] "Wilds: A benchmark of in-the-wild distribution shifts." International Conference on Machine Learning. PMLR, 2021.
>
> [Malinin et al. NeurIPS 2021] Malinin, Andrey, Neil Band, Yarin Gal, Mark Gales, Alexander Ganshin, German Chesnokov, Alexey Noskov et al. "Shifts: A Dataset of Real Distributional Shift Across Multiple Large-Scale Tasks." In Thirty-fifth Conference on Neural Information Processing Systems D&B Track. 2021.

---

### Official Review · Reviewer_R2F7 · 2022-07-27
**Reviews of the WilD-Time benchmark.**

**Rating:** 8
**Confidence:** 5

**Strengths:**

The authors have provided a valuable and diverse benchmark which focuses exclusively on distributional shift in time. This effect occurs in many services and practical applications, and therefore have broad implications for both research and for industrial practitioners. The work itself examines a broad range of applications, tasks and modalities. There are no negative ethical or social implications in this work.


**Weaknesses:**

The current work does not have any significant technical weaknesses. However, it would be interesting to examine time-shift in complex, structured prediction tasks, such as segmentation, autoregressive classification and regression. Additionally, it would be interesting to also consider Bayesian-and-related methods, such as Ensembles, and how much improvement in robustness they are able to provide. Finally, exploring the choice of model architecture and its impact on robustness.

However, there is a weakness in the support documentation. It is unclear how the dataset it to be supported, under what licenses they are shared / re-used, how the long-term support of the dataset will be carried out. The GitHub (which I know is being refactored) is inconsistent in how the data is to be obtained (either via notebooks or via google drive, etc...). This presents a challenge to accessing the data. These issues needs to be properly addressed for this paper to be accepted. I will be very happy to revise my assessment of the paper when the authors provide adequate documentation.

**Additional Feedback:**

None

**Clarity:**

The paper is clearly written and easy to follow. However, the giant table is _extremely_ hard to follow and spending time interpreting it greatly breaks the flow of the story and the paper. I would very strongly recommend breaking it down into far smaller tables which tell well-contained stories or choosing a different form which can convey the same information more clearly.

**Correctness:**

The dataset seems to have been partitioned sensibly, the evaluation and experiment design and also sound. The claim of the paper that ERM presents a strong baseline and all other approaches don't work well under distributional shift in time are consistent with previously published results regarding other forms of distributional shift.

**Documentation:**

The content, licensing, support and maintenance plans are unclear. I cannot find a proper discussion of it in the main paper, the appendix not the website. In the appendix the authors quote the licenses under which the datasets were originally received, but not necessary the license under which they are re-sharing the data. Additionally, they have quoted the incorrect license for the Shifts Weather dataset, which is CC BY NC SA 4.0, not apache 2.0 (which is for code, not data).

I strongly suggest the authors re-read the documentation requirement outlined in the 'submission instructions' here: https://neurips.cc/Conferences/2022/CallForDatasetsBenchmarks and make sure they cover all the explicitly mentioned points.

**Ethics:**

No concerns on my part.

**Relation To Prior Work:**

Relation to previous work is detailed well. However, I am unclear how the next Eval-Fix setting in the current dataset is different from the data partitioning of the Shifts Weather dataset, other than not covering different climate-zones. Is it the same?

**Summary And Contributions:**

The authors are examining distributional shifts in time, also known as concept drift. They construct a benchmark from a series of 7 existing datasets, develop two evaluation strategies and evaluation a range of methods augmentation, invariant learning and continuers learning methods for improving robustness on this benchmark. The authors show that no methods consistently outperform ERM, that augmentation can sometimes be detrimental, and that invariant learning does better than continuous / incremental learning approaches. The authors clearly show that current tools are insufficient for combatting distributional shift in time.

This is a good paper, but the documentation is lacking, and I will be happy to revise my assessment when the authors provide the necessary bits, as described below.

---

> ### Author Response · Authors · 2022-08-22
> **Response to Reviewer R2F7 (1/3)**
>
> Thank you for your valuable comments and feedback. We address your concerns below, and have revised our paper accordingly. Please let us know if you have any additional questions or concerns.
>
> **Q1**: The current work does not have any significant technical weaknesses. However, it would be interesting to examine time-shift in complex, structured prediction tasks, such as segmentation, autoregressive classification and regression.
>
> **A1**: We conduct an additional evaluation on the New York City Taxi dataset (https://www1.nyc.gov/site/tlc/about/tlc-trip-record-data.page). Given historical hourly traffic demand values, we aim to predict traffic demand in the next 12 hours. This problem can be formulated as an autoregressive regression task. Here, we adopt LSTNet [Lai et al. SIGIR 2018] as the backbone and use root mean squared error (RMSE) as the evaluation metric. We train the model on years 2011 - 2015 and evaluate the model on years 2016 - 2019. The average RMSE of the ID and OOD test sets are 0.0487 and 0.0509, respectively – a relatively small performance drop. One potential reason is that traffic demand is relatively stable in developed cities, such as New York City. Thus, we decided not to include this dataset in Wild-Time.
>
> However, we are actively seeking datasets and tasks that exhibit temporal distribution shifts and put this in our maintenance plan, and are happy to hear any suggestions you may have.
>
> ---
>
> **Q2**: Additionally, it would be interesting to also consider Bayesian-and-related methods, such as Ensembles, and how much improvement in robustness they are able to provide.
>
> **A2**: We added Eval-Fix results with Stochastic Weight Averaging (SWA) [Pavel et al. UAI 2018], an approximate Bayesian method which averages multiple parameter values along the trajectory of SGD. We used the hyperparameters previously tuned for ERM. The results are reported as SWA in the Eval-Fix setting. We reported the OOD average performance of ERM and SWA in Table Re-1 and put the full table in Table 2 of the revised paper.
>
> **Table Re-1**: Performance (OOD average performance) comparison between ERM and SWA under Eval-Fix setting. See the full table in Table 2 (Table 19 in Appendix) of the revised paper.
> |     |    Yearbook   |      FMoW     | MIMIC-Readmission | MIMIC-Mortality |
> |-----|:-------------:|:-------------:|:-----------------:|:---------------:|
> | ERM |  79.50 (6.23) |  **51.99 (0.37)** |    **58.51 (4.06)**   |   **69.74 (4.51)**  |
> | SWA |  **84.25 (3.06)** |  50.59 (0.46) |    41.89 (1.46)   |   63.60 (5.09)  |
> |     |    **Drug-BA**    | **Precipitation** |      **Huffpost**     |      **arXiv**      |
> | ERM | **0.357 (0.012)** |  **47.83 (0.58)** |    70.42 (1.15)   |   **45.94 (0.97)**  |
> | SWA | 0.355 (0.002) |  47.16 (1.10) |    **70.98 (0.05)**   |   44.36 (0.77)  |
>
> Compared with ERM, we observe that SWA does not outperform other approaches in most datasets under the Eval-Fix setting. This is as expected because while SWA has been shown to improve in-distribution generalization, it is not designed to handle OOD inputs or to incorporate the additional information inside timestamp metadata.

---

> > ### Author Response · Authors · 2022-08-22
> > **Response to Reviewer R2F7 (2/3)**
> >
> > **Q3**: Finally, exploring the choice of model architecture and its impact on robustness.
> >
> > **A3**: We’ve added an ablation in Appendix E.4 to assess the impact of model architecture on robustness. For FMoW-WildT, we ran baselines with both ResNet18 and ResNet50. For arXiv, we ran baselines with ALBERT and BERT backbones. We report the performance under Eval-Fix in Table Re-2, where the full table with detailed discussion is in Appendix E.4 of the revised paper.
> >
> > Table Re-2: Performance comparison under Eval-Fix setting w.r.t. Different backbones. See Appendix E.4 for full results and detailed discussion.
> >
> > |       |   Backbone  |      ERM     |  Fine-tuning |     LISA     |
> > |-------|:-----------:|:------------:|:------------:|:------------:|
> > | FMoW  |   ResNet18  | 46.30 (0.36) | 41.53 (0.30) | 46.70 (0.20) |
> > |       |   ResNet50  | 51.56 (1.25) | 47.73 (2.21) | 51.40 (1.45) |
> > |       | DenseNet101 | 51.99 (0.37) | 45.77 (0.53) | 48.76 (0.48) |
> > | arXiv |  DistilBERT | 45.94 (0.97) | 50.31 (0.39) | 47.82 (0.47) |
> > |       |     BERT    | 47.51 (1.20) | 50.99 (0.52) | 49.05 (1.01) |
> > |       |    ALBERT   | 45.25 (0.65) | 49.76 (0.69) | 46.01 (0.52) |
> >
> > The new results with different backbones are consistent with our prior findings, i.e., both invariant learning and continual learning approaches do not make models more robust to temporal distribution shift, even with different backbones. We would like to note that while there are many promising directions for improving temporal robustness, such as new model architectures and pretraining, in Wild-Time, we focus on evaluating the learning algorithm itself.
> >
> > ---
> >
> > **Q4**: The paper is clearly written and easy to follow. However, the giant table is extremely hard to follow and spending time interpreting it greatly breaks the flow of the story and the paper. I would very strongly recommend breaking it down into far smaller tables which tell well-contained stories or choosing a different form which can convey the same information more clearly.
> >
> > **A4**: We replaced Table 1 in the initial version with smaller tables (Table 1 & 2 in the revised version) that convey self-contained stories. Specifically, we made the following changes:
> >
> > - We split ID and OOD performance, and use one subsection (Section 5.2) to discuss the performance drop between ID and OOD datasets with the analysis of task difficulty.
> > - We remove the standard deviations in Table 1 and include the full table in Appendix (Table 19).
> >
> > ---
> >
> > **Q5**: Relation to previous work is detailed well. However, I am unclear how the next Eval-Fix setting in the current dataset is different from the data partitioning of the Shifts Weather dataset, other than not covering different climate-zones. Is it the same?
> >
> > **A5**: Yes, other than not covering different climate-zones, it is the same.
> >
> > ---
> >
> > **Q6**: The content, licensing, support and maintenance plans are unclear. I cannot find a proper discussion of it in the main paper, the appendix not the website. In the appendix the authors quote the licenses under which the datasets were originally received, but not necessary the license under which they are re-sharing the data. Additionally, they have quoted the incorrect license for the Shifts Weather dataset, which is CC BY NC SA 4.0, not apache 2.0 (which is for code, not data)
> >
> > **A6**: We apologize for the mistake and confusion. We’ve fixed all issues and made the following changes to our support documentation.
> > - We updated the Wild-Time GitHub with detailed instructions for downloading the datasets and running all baselines.
> > - We added a script that users can run to download all Wild-Time datasets.
> >   - For the MIMIC-WildT dataset, users must first get credentialed access to the original MIMIC-IV dataset due to healthcare data use policies. This is a relatively straightforward and quick process, and we included instructions for how to do so.
> > - We added Section G in the Appendix, where we state the licenses under which we are re-sharing the data, along with the hosting, support, maintenance plan, and author statement. Specifically, our maintenance plan includes:
> >   - Monitoring the GitHub page to address any questions or issues with Wild-Time;
> >   - Including more baseline methods and diverse datasets and updating tasks as needed;
> >   - Releasing an open-source, PyTorch-based Wild-Time package that automates data loading, evaluation, and baseline model training with a simple user interface;
> >   - Maintaining a website for the Wild-Time benchmark, which will be used to (1) announce version updates and (2) host a leaderboard to track state-of-the-art algorithms and model backbones.
> >
> > We are actively working on code refactoring and building APIs for the Wild-Time package. We plan to release the final version in early September.

---

> > > ### Author Response · Authors · 2022-08-22
> > > **Response to Reviewer R2F7 (3/3)**
> > >
> > > **References**
> > >
> > > [Lai et al. SIGIR 2018] Lai, Guokun, Wei-Cheng Chang, Yiming Yang, and Hanxiao Liu. "Modeling long-and short-term temporal patterns with deep neural networks." In The 41st international ACM SIGIR conference on research & development in information retrieval, pp. 95-104. 2018.
> > >
> > > [Pavel et al. UAI 2018] Izmailov, Pavel, Dmitrii Podoprikhin, Timur Garipov, Dmitry Vetrov, and Andrew Gordon Wilson. "Averaging weights leads to wider optima and better generalization." UAI 2018.

---

### Official Review · Reviewer_vzrG · 2022-07-27
**This work presents a benchmark of out of distribution shift generalization over the time**

**Rating:** 5
**Confidence:** 4
**Clarity:** Yes

**Strengths:**

- This paper is motivated and clearly presented

- This work provides a set of curated Time-OOD datasets, and results of conventional results. It would be a suitable starting point for the researcher interested to work on.


**Weaknesses:**


While the realistic temporal issue needs to be addressed, there are some other issues to consider:


- The temporal shift distribution curated in this paper are often intermixed with other types of distribution shifts. For example, in the FMOW dataset, the authors directly partitioned the dataset by year, ignoring location consistency across sets, which would result in performance degradation possibly due to shifts caused by locations. In the dataset Yearbook, whether the male-to-female division is consistent between time periods 1930-1970 and 1971-2013 also affects the performance difference. Thus an important question is how to ensure that the performance degradation is purely due to time distribution shift and not due to inconsistencies of other domains.

- The purpose of this paper is to propose datasets with time distribution shift, however, some settings are not quite consistent with the reality. For example, under the main Eval-Fix setting, the time span between train and OOD test for many sub-datasets is huge, for example, in task 1, the training set is on average 20 years older than the test set, and this setting may be uncommon in the standard  ML development pipeline. What we do know is that model updates are frequent.  It would be helpful if you could resolve my doubts about this, namely that some of the proposed dataset settings are not necessarily reality-oriented.


**Additional Feedback:**

No

**Correctness:**

Yes, it sounds good but I still have some considerations, please see the Weaknesses part.

**Documentation:**

The authors released a preliminary version of Wild-Time.

**Relation To Prior Work:**

Yes

**Summary And Contributions:**

This paper advocate dealing with the issue of distribution shift over time for robust model performance. The author curated 7 datasets from existing datasets that reflect realistic temporal distribution shifts.  Conventional domain generalization methods, as well as continual learning-based methods, are benchmarked under two different settings (Eval-Fix, Eval-Steam). The experiments demonstrate that the benchmarked methods are limited in dealing with temporal distribution shifts across all proposed settings. This work hopes to assist the community in tackling the problem of temporal distribution shift by proving benchmarking datasets and baseline models.

---

> ### Author Response · Authors · 2022-08-22
> **Response to Reviewer vzrG (1/2)**
>
> Thank you for your helpful and insightful feedback. We address your concerns below, and have revised our paper in response to your comments. Please let us know if our response addresses all of your considerations.
>
> **Q1**: For example, in the FMOW dataset, the authors directly partitioned the dataset by year, ignoring location consistency across sets, which would result in performance degradation possibly due to shifts caused by locations.
>
> **A1**: We’ve re-preprocessed the FMoW-WildT dataset to follow the FMoW-WILDS split, which accounts for geographic location. Concretely, we follow the splits in WILDS [Koh et al. ICML 2021] to partition each year's data into train/test for FMoW-WildT. For $2002 - 2012$, we combine the FMoW-WILDS Training (ID) and Validation (ID) splits for training, and allocate the Test (ID) split to testing. For $2013 - 2015$, we allocate 90\% of the data from each year in the Validation (OOD) split to train, and the remaining 10\% to test. For $2016-2017$, we allocate 90\% of the data from each year in the Test (OOD) split to train, and the remaining 10\% to validation. Our fixed time split (Eval-Fix) uses the first $14$ years ($2002 - 2015$) for training, and the remaining $2$ years ($2016 - 2017$) for testing. We report the new FMoW-WildT baseline results in Table Re-1 (Eval-Fix). The results of Eval-Stream are reported in Appendix D of the revised paper.
>
> **Table Re-1**: Results of FMoW with new split under Eval-Fix setting.
>
> |             | FMoW (Accuracy $\uparrow$) |              |              |
> |-------------|:--------------------------:|:------------:|:------------:|
> |             |           ID Avg.          |   OOD Avg.   |   OOD Worst  |
> | Fine-tuning |        52.56 (0.18)        | 45.77 (0.53) | 43.21 (0.85) |
> | EWC         |        52.43 (0.62)        | 45.60 (0.28) | 43.13 (0.50) |
> | SI          |        52.84 (0.30)        | 44.87 (0.73) | 42.97 (1.15) |
> | A-GEM       |        52.63 (0.56)        | 45.21 (0.20) | 42.49 (0.70) |
> | ERM         |        60.88 (0.34)        | **51.99 (0.37)** | **48.79 (0.49)** |
> | GroupDRO-T  |        45.98 (0.08)        | 37.61 (1.16) | 34.41 (1.39) |
> | Mixup       |        58.46 (0.43)        | 49.82 (0.19) | 45.58 (0.31) |
> | LISA        |        55.97 (0.29)        | 48.76 (0.48) | 45.41 (0.21) |
> | CORAL-T     |        56.02 (0.38)        | 47.34 (0.09) | 44.04 (0.46) |
> | IRM-T       |        45.86 (1.08)        | 38.73 (1.67) | 34.93 (1.88) |
>
> The above results corroborate our main findings that both invariant learning approaches and continual learning approaches do not make models more robust to temporal distribution shift.
>
> ---
>
> **Q2**: The temporal shifts curated in the Wild-Time benchmark are often intermixed with other types of distribution shifts. In the dataset Yearbook, whether the male-to-female division is consistent between time periods 1930-1970 and 1971-2013 also affects the performance difference. Thus an important question is how to ensure that the performance degradation is purely due to time distribution shift and not due to inconsistencies of other domains.
>
> **A2**: We’ve added label distribution plots for all Wild-Time datasets in Appendix E.1. In all Wild-Time datasets, we observe that the label distribution changes over time. With Wild-Time, we strive to capture temporal distribution shifts that occur in the real world. These natural distribution shifts may be a mixture of different types of shifts over time, including label shifts. We’ve clarified the definition of temporal distribution shift in Line 45-47 of the revised paper. We show that existing approaches (ERM, invariant learning, continual learning, self-supervised learning, ensemble methods) fail to close the gap between OOD and ID performance, highlighting the need for more powerful algorithms.
>
> In addition, we included an analysis in Appendix E.1 of our initial submission (also in the revised version), where we verify that the performance gap between ID and OOD timestamps are not caused by the difficulty of examples from OOD timestamps. Specifically, to measure task difficulty, we use two kinds of data splits -- standard split and mixed split. In the standard split, the model is trained on timestamps before the split time and then evaluated on examples from future timestamps. In the mixed split, the training data is merged from all timestamps, and the model is evaluated on the original OOD examples. We train ERM on both splits and observe large performance gaps between standard split and mixed split on all Wild-Time tasks. This verifies that the performance gaps between ID and OOD are not caused by the difficulty of the timestamps.

---

> > ### Author Response · Authors · 2022-08-22
> > **Response to Reviewer vzrG (2/2)**
> >
> > **Q3**: The purpose of this paper is to propose datasets with time distribution shift, however, some settings are not quite consistent with the reality. For example, under the main Eval-Fix setting, the time span between train and OOD test for many sub-datasets is huge, for example, in task 1, the training set is on average 20 years older than the test set, and this setting may be uncommon in the standard ML development pipeline. What we do know is that model updates are frequent. It would be helpful if you could resolve my doubts about this, namely that some of the proposed dataset settings are not necessarily reality-oriented.
> >
> > **A3**: We acknowledge that the Eval-Fix setting is less realistic in the Yearbook task, though the large time gap was used in some ML applications, e.g., water temperature prediction [Chen et al. KDD 2022]. Thus, we provided a secondary evaluation strategy, Eval-Stream (Lines 110-114 of the revised version), which better reflects ML development pipelines. In Eval-Stream, we update the model at each timestamp and evaluate over the next $K$ timestamps. We mention the applicability of Eval-Stream to ML development pipelines in Lines 112-113 of our revised version.
> >
> > We chose to make Eval-Fix the primary evaluation setting for the following reasons, which we discussed in Lines 68-75 (Section 1: Introduction) of our initial submission:
> > 1. Eval-Fix serves as a unittest to quickly verify the performance of new approaches, as it adopts a single split to get the training and test time periods. We believe that this makes our benchmark more user-friendly, while still encapsulating a slice of the ML development pipeline/continual learning setting.
> > 2. Eval-Fix is more usable by the general machine learning community, and is consistent with other distribution shift benchmarks, such as WILDS [Koh et al. ICML 2021] and SHIFTS [Malinin et al. NeurIPS 2021].
> >
> > ---
> >
> > **Reference**
> >
> > [Koh et al. ICML 2021] Koh, Pang Wei, Shiori Sagawa, Henrik Marklund, Sang Michael Xie, Marvin Zhang, Akshay Balsubramani, Weihua Hu et al. "Wilds: A benchmark of in-the-wild distribution shifts." In International Conference on Machine Learning, pp. 5637-5664. PMLR, 2021.
> >
> > [Malinin et al. NeurIPS 2021] Malinin, Andrey, Neil Band, Yarin Gal, Mark Gales, Alexander Ganshin, German Chesnokov, Alexey Noskov et al. "Shifts: A Dataset of Real Distributional Shift Across Multiple Large-Scale Tasks." In Thirty-fifth Conference on Neural Information Processing Systems Datasets and Benchmarks Track (Round 2). 2021.
> >
> > [Chen et al. KDD 2022] Chen, Shengyu, Jacob A. Zwart, and Xiaowei Jia. "Physics-Guided Graph Meta Learning for Predicting Water Temperature and Streamflow in Stream Networks." In Proceedings of the 28th ACM SIGKDD Conference on Knowledge Discovery and Data Mining, pp. 2752-2761. 2022.

---

### Official Review · Reviewer_QHWk · 2022-07-27
**Extensive temporal OOD benchmark but can be improved**

**Rating:** 5
**Confidence:** 4
**Correctness:** Mostly correct. A few statements have…
**Clarity:** The paper is generally easy to follow…

**Strengths:**

1. Out-of-distribution problem is an essential field increasingly gaining attention, and temporal out-of-distribution datasets are not discussed much by existing work, therefore collecting datasets for this field is meaningful contribution.
2. Methods specifically designed for temporal distribution shifts are majorly underdeveloped. The paper evaluate performances of different continual learning and invariant learning methods on their datasets, providing useful data to guide the development of future temporal OOD methods.
3. The paper is generally easy to follow and understand.

**Weaknesses:**

1. My major concern is in the data split. For the Eval-Stream setting, which should be closer to the acknowledged domain design in OOD datasets and models can incrementally learn between time steps, the split seems questionable. For example the authors "allocate 10% of the data at each timestep for test, and the rest for training. For OOD testing, all samples in each year are used", this means that the ID training has 90% data, ID test 10% but OOD test 100% data. I suppose this means that they use the data other than the current tilmestep to do OOD test, or it would not fit my understanding of OOD test setting. But this unbalanced ID/OOD train/test split would make the results incomparable with Eval-Fix. And I'm not sure if this would negatively affect the best performance of incrementally learning strategies. The time steps could be designed as domains like general OOD datasets.
2. Selection of methods and models lack motivation reasoning. The paper evaluated algorithms on different model backbones but give no comparison of other model backbones. Since the conclusion includes that no existing continual/invariant learning approach generally outperforms ERM on temporal distribution shifts, careful selection and reasoning of methods and models should be vital. Also some important OOD-related methods are not compared or discussed. Frequently used methods in OOD generalization recently, such as mixup (Zhang, Hongyi, et al. "mixup: Beyond empirical risk minimization." arXiv preprint arXiv:1710.09412 (2017)), is not compared. Also, image and text data have no image/text-specified OOD-related methods compared.
3. Some experimental details are not provided. The paper lack some necessary data to prove some claims convincing. For example, algorithms such as IRM has hyperparameters that can have a wide range of value selection, the paper did not provide the fine-tuning process or the reason for parameter selection, and the hyperparameters can largely affect the performance and observations.
4. The GitHub code documentation are not organized much. As datasets are primarily for community use, it should be easily usable and the documentation should be user-friendly. The GitHub repo has few starting instructions, so it is difficult to use, for example, the modules for the datasets specifically. I would suggest the GitHub documentation be actually maintained.

**Additional Feedback:**

The suggestions are provided along with the comments in the Weaknesses part.

**Documentation:**

The licensing and dataset accessibility is provided. The GitHub documentation page should include more details, such as instructions on how to use the modules for the datasets.

**Ethics:**

No ethical concerns.

**Relation To Prior Work:**

Relation to previous work is discussed in the paper, the main point is that temporal out-of-distribution datasets are not discussed much by existing work.

**Summary And Contributions:**

The Wild-Time collects 7 temporal out-of-distribution datasets, among which 2 are image datasets, 1 tabular dataset, 1 graph molecular dataset, and 3 text datasets. The paper aims at time domains and use/define temporal distribution shift of multiple time steps for each dataset. They test the in-distribution and out-of-distribution performance on multiple methods, including comparison of different continual learning and invariant learning algorithms. The paper also discusses the observations from the experimental results and analyze the comparisons to give brief conclusions, that no existing continual/invariant learning approach generally outperforms ERM on temporal distribution shifts.

---

> ### Author Response · Authors · 2022-08-22
> **Response to Reviewer QHWk (1/3)**
>
> Thank you for your valuable comments. We have revised our paper based on your feedback, and believe our paper is stronger as a result. We describe the revisions we made below. Please let us know if our response addresses all of your concerns.
>
>
> **Q1**: My major concern is in the data split. For the Eval-Stream setting, … the authors "allocate 10% of the data at each timestep for test, and the rest for training. For OOD testing, all samples in each year are used", this means that the ID training has 90% data, ID test 10% but OOD test 100% data. I suppose this means that they use the data other than the current tilmestep to do OOD test, or it would not fit my understanding of OOD test setting. But this unbalanced ID/OOD train/test split would make the results incomparable with Eval-Fix. And I'm not sure if this would negatively affect the best performance of incrementally learning strategies. The time steps could be designed as domains like general OOD datasets.
>
> **A1**: We apologize for the confusion. Our Eval-Fix and Eval-Stream settings are comparable. Suppose we have $T$ timestamps. At each timestamp, we randomly sample 90% of the examples for training, and allocate the remaining 10% examples for ID evaluation.
>
> In Eval-Fix, we have a split timestamp $t_{s}$. The ID timestamps are $t < t_{s}$, and the OOD timestamps are $t \geq t_{s}$. The training set consists of all training examples from the ID timestamps $t < t_{s}$. The ID validation set consists of all validation examples from the ID timestamps $t < t_{s}$.  All examples in all test timestamps $t \geq t_{s}$ are used as the OOD test set.
>
> In Eval-Stream, at each evaluation timestamp, we evaluate across the next $K$ timestamps. Specifically, at each timestamp $t \in [1, …, T]$, we evaluate our model across the timestamps $t + 1, \dots, t + K$.
>
> Hence, Eval-Fix can be viewed as a single timestamp evaluation within Eval-Stream, where we evaluate only at $t_{s}$ and set $K = T - t_{s}$.
>
> We’ve clarified this in Appendix C.2 of our revised paper. We also added Figure 2 to illustrate how we split the data under both Eval-Fix and Eval-Stream settings.
>
> ---
>
> **Q2**: Selection of methods and models lack motivation reasoning. The paper evaluated algorithms on different model backbones but give no comparison of other model backbones. Since the conclusion includes that no existing continual/invariant learning approach generally outperforms ERM on temporal distribution shifts, careful selection and reasoning of methods and models should be vital.
>
> **A2**: For each dataset, we use the same backbone for all baselines. The choice of backbone is either from the original paper (e.g., DenseNet101 for FMoW [Koh et al. ICML 2021], DeepDTA for Drug-BA [Huang et al. NeurIPS 2021]) or a commonly-used backbone (e.g., DistilBERT for text classification in arXiv and Huffpost). Regarding the choice of baselines, we selected representative methods for each category of approaches (e.g., EWC and A-GEM in continual learning, CORAL, IRM and LISA in invariant learning). We’ve clarified the choice of baseline methods and model backbones in the first paragraph of Section 4 and in “Hyperparameter Settings” (Section 5.1) of the revised version, respectively.
>
> Furthermore, we conducted new experiments with different backbones for an image dataset (FMoW) and text dataset (arXiv). Specifically, we use ResNet18, ResNet50 for FMoW, and BERT and ALBERT for arXiv. We report the results of ERM and two representative approaches -- LISA and Fine-tuning under the Eval-Fix setting in Table Re-1.
>
> **Table Re-1**: Performance comparison w.r.t. Different backbones. See Appendix E.4 for detailed discussion.
>
> |       |   Backbone  |      ERM     |  Fine-tuning |     LISA     |
> |-------|:-----------:|:------------:|:------------:|:------------:|
> | FMoW  |   ResNet18  | 46.30 (0.36) | 41.53 (0.30) | 46.70 (0.20) |
> |       |   ResNet50  | 51.56 (1.25) | 47.73 (2.21) | 51.40 (1.45) |
> |       | DenseNet101 | 51.99 (0.37) | 45.77 (0.53) | 48.76 (0.48) |
> | arXiv |  DistilBERT | 45.94 (0.97) | 50.31 (0.39) | 47.82 (0.47) |
> |       |     BERT    | 47.51 (1.20) | 50.99 (0.52) | 49.05 (1.01) |
> |       |    ALBERT   | 45.25 (0.65) | 49.76 (0.69) | 46.01 (0.52) |
>
>
> The new results with different backbones are consistent with our prior findings, i.e., both invariant learning and continual learning approaches do not make models more robust to temporal distribution shift, even with different backbones. We’ve added these discussions in Appendix E.4.

---

> > ### Author Response · Authors · 2022-08-22
> > **Response to Reviewer QHWk (2/3)**
> >
> > **Q3**: Also some important OOD-related methods are not compared or discussed. Frequently used methods in OOD generalization recently, such as mixup (Zhang, Hongyi, et al. "mixup: Beyond empirical risk minimization." arXiv preprint arXiv:1710.09412 (2017)), is not compared.
> >
> > **A3**: We already evaluated mixup on all Wild-Time datasets and reported the results in Appendix E.3 and Table 13 (Appendix) of our initial submission. Here, we observe that in most datasets (e.g. Yearbook, FMoW, MIMIC-Mortality, Drug-BA, Precipitation, arXiv), vanilla mixup worsens out-of-distribution generalization compared to ERM. We’ve moved the mixup results to the main paper (Table 2) in the revised version.
> >
> > ---
> >
> > **Q4**: Also, image and text data have no image/text-specified OOD-related methods compared.
> >
> > **A4**: To evaluate image-related OOD methods, we implemented two self-supervised learning approaches: SimCLR [Chen et al. ICML 2020] and SwAV [Caron et al. NeurIPS 2020] that have been shown to improve the out-of-distribution robustness [Ji et al. Preprint 2022; Shen et al. ICML 2022]. Following [Chen et al. ICML 2020], we first leverage the aforementioned self-supervised learning approaches to learn representations, and then we use the same data to fine-tune the classifier. We report these results in Table Re-1, and also report the performance of ERM for comparison.
> >
> > **Table Re-2**: Performance (accuracy with standard deviation) of self-supervised learning approaches on image classification.
> >
> > |        | Yearbook (Accuracy $\uparrow$) |              | FMoW (Accuracy $\uparrow$) |              |
> > |--------|:------------------------------:|:------------:|:--------------------------:|:------------:|
> > |        |              Avg.              |     Worst    |            Avg.            |     Worst    |
> > | ERM    |          **79.50 (6.23)**          | **63.09 (5.15)** |        **51.99 (0.37)**        | **48.79 (0.49)** |
> > | SimCLR |          78.59 (2.72)          | 60.15 (3.48) |        42.91 (0.40)        | 39.54 (0.67) |
> > | SwAV   |          78.38 (1.86)          | 60.73 (1.08) |        49.53 (0.27)        | 46.31 (0.58) |
> >
> >
> > According to the performance, we observe that these self-supervised learning approaches do not show significant benefits on image data, corroborating our main conclusion. We add these results and discussions in Table 2 of the revised paper.
> >
> > We were not able to find a practical, text-specific, OOD method. We found that most text-related OOD approaches are existing continual learning methods (e.g., continual fine-tuning, EWC, and buffer replay [Lin et al. ACL 2022]). We are happy to hear any suggestions you may have regarding such approaches and can conduct further experiments.
> >
> > ---
> >
> > **Q5**: Some experimental details are not provided. The paper lack some necessary data to prove some claims convincing. For example, algorithms such as IRM has hyperparameters that can have a wide range of value selection, the paper did not provide the fine-tuning process or the reason for parameter selection, and the hyperparameters can largely affect the performance and observations.
> >
> > **A5**: We tuned the hyperparameters of IRM and other approaches via cross-validation with grid search, which is widely used for hyperparameter tuning in the machine learning community. Specifically, in Eval-Fix, instead of using an in-distribution validation set, we hold out 10% of the data of each training time step (20% for Drug-BA, MIMIC-Readmission, and MIMIC-Mortality) to construct the out-of-distribution validation set. Here, we use examples from the remaining 90\% of the data time steps to train the model and evaluate the out-of-distribution performance on the validation set for hyperparameter tuning. We repeat this process several times via cross-validation. After tuning all hyperparameters, we use the entire training set to train the model.
> >
> > ---
> >
> > **Q6**: The GitHub code documentation are not organized much. As datasets are primarily for community use, it should be easily usable and the documentation should be user-friendly. The GitHub repo has few starting instructions, so it is difficult to use, for example, the modules for the datasets specifically. I would suggest the GitHub documentation be actually maintained.
> >
> > **A6**: We have updated the GitHub documentation with more details. To make the GitHub repo more user-friendly, we added the following to the code documentation:
> > 1. A script that users can run to download all Wild-Time datasets;
> > 2. Detailed instructions for how to run all baselines on the Wild-Time dataset;.
> > 3. Licenses under which we are sharing the Wild-Time benchmark.
> >
> > We also added a maintenance plan to Appendix G of our paper, which discusses how we plan to support the Wild-Time repo.

---

> > > ### Author Response · Authors · 2022-08-22
> > > **Response to Reviewer QHWk (3/3)**
> > >
> > > **References**
> > >
> > > [Koh et al. ICML 2021] Koh, Pang Wei, Shiori Sagawa, Henrik Marklund, Sang Michael Xie, Marvin Zhang, Akshay Balsubramani, Weihua Hu et al. "Wilds: A benchmark of in-the-wild distribution shifts." In International Conference on Machine Learning, pp. 5637-5664. PMLR, 2021.
> > >
> > > [Huang et al. NeurIPS 2021] Huang, Kexin, Tianfan Fu, Wenhao Gao, Yue Zhao, Yusuf H. Roohani, Jure Leskovec, Connor W. Coley, Cao Xiao, Jimeng Sun, and Marinka Zitnik. "Therapeutics Data Commons: Machine Learning Datasets and Tasks for Drug Discovery and Development." In Thirty-fifth Conference on Neural Information Processing Systems Datasets and Benchmarks Track (Round 1). 2021.
> > >
> > > [Chen et al. ICML 2020] Chen, Ting, Simon Kornblith, Mohammad Norouzi, and Geoffrey Hinton. "A simple framework for contrastive learning of visual representations." In International conference on machine learning, pp. 1597-1607. PMLR, 2020.
> > >
> > > [Caron et al. NeurIPS 2020] Caron, Mathilde, Ishan Misra, Julien Mairal, Priya Goyal, Piotr Bojanowski, and Armand Joulin. "Unsupervised learning of visual features by contrasting cluster assignments." Advances in Neural Information Processing Systems 33 (2020): 9912-9924.
> > >
> > > [Ji et al. Preprint 2022] Ji, Wenlong, Zhun Deng, Ryumei Nakada, James Zou, and Linjun Zhang. "The power of contrast for feature learning: A theoretical analysis." arXiv preprint arXiv:2110.02473 (2021).
> > >
> > > [Shen et al. ICML 2022] Shen, Kendrick, Robbie M. Jones, Ananya Kumar, Sang Michael Xie, Jeff Z. HaoChen, Tengyu Ma, and Percy Liang. "Connect, not collapse: Explaining contrastive learning for unsupervised domain adaptation." In International Conference on Machine Learning, pp. 19847-19878. PMLR, 2022.
> > >
> > > [Lin et al. ACL 2022] Lin, Bill Yuchen, Sida I. Wang, Xi Lin, Robin Jia, Lin Xiao, Xiang Ren, and Scott Yih. "On Continual Model Refinement in Out-of-Distribution Data Streams." In Proceedings of the 60th Annual Meeting of the Association for Computational Linguistics (Volume 1: Long Papers), pp. 3128-3139. 2022.

---

> ### Comment · Reviewer_QHWk · 2022-08-28
> **Many thanks to the work and detailed explanations**
>
> Thanks for the detailed explanations for my concerns. I appreciate the work done by the authors for the revision. Some of my concerns are addressed, however I still have doubts in several aspects.
>
> 1. The design of Eval-Stream setting is now clear to me. But given it has little analysis and observations results, only "invariant learning approaches performs slightly better than continual learning approaches in most tasks" in the appendix, I do not quite see the significance of this setting. Does this design have certain meaning? If it's for "sequential evaluation methodology in continual learning and aims to accelerate continual learning research", why are continual learning approaches not showing any improvements? Is there an explanation or further analysis? If this setting is to be kept, should its significance be justified in someway?
>
> 2. The design of settings are not particularly novel, and most datasets are adopted from existing works. Though the paper has a good motivation and makes a comprehensive benchmark on temporal OOD, I feel the significance of the contribution not particularly stimulating.
>
> 3. Related works can be further extended. Many other works regarding distribution shifts and OOD should be addressed in Relation to Distribution Shift Benchmarks. For example OoD-Bench[1] and NICO[2] has been around for a while and seems related.
>
> To summarize, I do appreciate the amount of work of the authors to curate the benchmark, and this work can have a high potential to impact in this field. But I feel the paper needs further revisions to be convincing enough for me. Thus I still lean to keep my score.
>
> [1]Ye, Nanyang, Kaican Li, Lanqing Hong, Haoyue Bai, Yiting Chen, Fengwei Zhou, and Zhenguo Li. "Ood-bench: Benchmarking and understanding out-of-distribution generalization datasets and algorithms." arXiv preprint arXiv:2106.03721 (2021).
> [2]He, Yue, Zheyan Shen, and Peng Cui. "Towards non-iid image classification: A dataset and baselines." Pattern Recognition 110 (2021): 107383.

---

> > ### Author Response · Authors · 2022-08-28
> > **Response to Additional Comments of Reviewer QHWk (1/2)**
> >
> > Thank you for your additional thoughtful comments. We address your concerns below. We also made some additional paper revisions in response to your comments. Please let us know if we addressed all of your concerns.
> >
> > **Q1**: The design of Eval-Stream setting is now clear to me. But given it has little analysis and observations results, only "invariant learning approaches performs slightly better than continual learning approaches in most tasks" in the appendix, I do not quite see the significance of this setting. Does this design have certain meaning? If it's for "sequential evaluation methodology in continual learning and aims to accelerate continual learning research", why are continual learning approaches not showing any improvements? Is there an explanation or further analysis? If this setting is to be kept, should its significance be justified in someway?
> >
> > **A1**: Yes, the design of Eval-Stream serves two purposes. First, Eval-Stream is geared towards continual learning and lifelong learning research. Second, Eval-Stream is a realistic setting for evaluating real-world ML systems, where data and labels often arrive asynchronously, requiring a model that can forecast into the near future.
> > Existing continual learning approaches fail to improve over other baselines in the Eval-Stream setting for the following two reasons:
> > - Most existing continual learning approaches focus on backward transfer, i.e., catastrophic forgetting. In Wild-Time, we focus on forward transfer. We consider the setting in which we evaluate performance on future timestamps, i.e., “temporal robustness.”
> > - For continual learning approaches that also focus on forward transfer (e.g., A-GEM), most of these approaches only show improvements on manually delineated sets of tasks with artificial temporal variations (e.g., Split CUB, Split CIFAR), but are not evaluated on benchmarks with natural temporal distribution shifts, such as Wild-Time. Analogously, we note that invariant learning approaches show improvements in artificial datasets (e.g., ColoredMNIST, Waterbirds [Sagawa et al. ICLR 2020]), but fail to outperform ERM in benchmarks with natural distribution shifts, e.g., WILDS [Koh et al. ICML 2021].
> > We’ve added the above explainations in Appendix D of the revised version.
> > These findings demonstrate that training a continual learning model that is robust to temporal distribution shifts remains a significant open challenge. We hope that Wild-Time can accelerate research on robust methods for tackling real-world temporal shifts.
> >
> > ---
> >
> > **Q2**: The design of settings are not particularly novel, and most datasets are adopted from existing works. Though the paper has a good motivation and makes a comprehensive benchmark on temporal OOD, I feel the significance of the contribution not particularly stimulating.
> >
> > **A2**: Although most datasets in Wild-Time are adopted from existing datasets, curating a new benchmark requires substantial design decisions, such as (1) which datasets to include and what criteria to use to determine if a dataset was suitable for our benchmark, and (2) how to frame and measure temporal distribution shifts.
> >
> > - For (1), we ensured that all datasets in Wild-Time exhibit substantial performance gaps between training and test timestamps. We conducted further experiments in Appendix E.1 to ensure that the performance drop was due to temporal shifts, and not due to the difficulty of the timestamps themselves. In addition to the clear performance gaps, we ensured that both training and test timestamps have sufficient examples under the Eval-Fix evaluation setting. In Appendix A, we discuss how we modify each dataset from its original version.
> >
> > - For (2), we introduce a new problem setting, temporal robustness, in which we focus on robustness into the future. For this new setting, we propose two evaluation strategies, Eval-Fix and Eval-Stream. Temporal robustness merges both continual/lifelong learning and out-of-distribution robustness research. Hence, we standardized evaluation in a way that is reflective of both continual learning/real-world streaming settings (Eval-Stream), but also easy for broader ML researchers to quickly develop on (Eval-Fix).
> > We believe that there can be substantial value in benchmarks that build on existing datasets. In particular, we believe that Wild-Time can contribute to the community’s understanding of how to frame and handle natural temporal distribution shifts, which are underrepresented in existing benchmarks.
> >
> > ---
> >
> > **Q3**: Related works can be further extended. Many other works regarding distribution shifts and OOD should be addressed in Relation to Distribution Shift Benchmarks. For example OoD-Bench[1] and NICO[2] has been around for a while and seems related.
> >
> > **A3**: Thanks for pointing out these related works. We’ve added NICO in Line 371 of the related work section and OoD-Bench in Line 57 of the introduction in our revised version.

---

> > > ### Author Response · Authors · 2022-08-28
> > > **Response to Additional Comments of Reviewer QHWk (2/2)**
> > >
> > > **References**
> > >
> > > [Sagawa et al. ICLR 2020] Sagawa, Shiori, Pang Wei Koh, Tatsunori B. Hashimoto, and Percy Liang. "Distributionally Robust Neural Networks." In International Conference on Learning Representations. 2020.
> > >
> > > [Koh et al. ICML 2022] Koh, Pang Wei, Shiori Sagawa, Henrik Marklund, Sang Michael Xie, Marvin Zhang, Akshay Balsubramani, Weihua Hu et al. "Wilds: A benchmark of in-the-wild distribution shifts." In International Conference on Machine Learning, pp. 5637-5664. PMLR, 2021.

---

> > > > ### Author Response · Authors · 2022-08-29
> > > > **We would like to hear back from Reviewer QHWk about our additional response**
> > > >
> > > > Dear Reviewer QHWk,
> > > >
> > > > Thank you again for your valuable additional comments. Since the end of the discussion period is approaching, we would like to follow up to see if our response addresses your additional comments. We would really appreciate it if you could check our additional response at your earliest convenience. Many thanks!

---

> > > > ### Comment · Reviewer_QHWk · 2022-08-29
> > > > **Thanks for the explanation. I look forward to further temporal-shift-specific OOD improvements of this work**
> > > >
> > > > Many thanks for the detailed explanations for my concerns. I appreciate the analysis and revision. I should say my concerns are somehow addressed.
> > > > Regarding the current design, this work does not appear to have any significant technical flaws and could be a good initial testbed for temporal OOD problems.
> > > >
> > > > However, since a temporal OOD benchmark should focus on temporal-shift-specific discussions, I put the design of Eval-Stream setting in a high position, given that Eval-Stream carries more uniqueness and challenges of temporal shifts while Eval-Fix resembles general OOD processing. I understand the intention of the Eval-Stream setting design but is not fully convinced that it well describes realistic temporal shifts and presents continual learning, with the current experiments and analysis. The proposed temporal robustness is an inspiring contribution; likewise, I look forward to see more temporal specific OOD contribution as a major  improvement.

---

> > > > > ### Author Response · Authors · 2022-08-31
> > > > > **Thank you for your response and we plan to investigate more potential evaluation strategies for the streaming setting.**
> > > > >
> > > > > Dear Reviewer QHWk,
> > > > >
> > > > > Thank you for your quick response. Sorry for the late reply, the discussion period ended two days before but we just found we can add comments today. We're very happy that our response addresses some of your questions. We plan to move some results of Eval-Stream from Appendix to the main paper. Sorry we can not edit our paper now because the revising system is closed. We also plan to investigate more potential evaluation strategies for the streaming setting in the future. Thanks again for your valuable comments to help us improve our paper.

---

### Author Response · Authors · 2022-08-22
**Summary of Paper Revisions (Updated 08/29)**

We sincerely thank all reviewers for their insightful and constructive feedback. We have revised our paper according to your comments and uploaded the new version. All paper edits are highlighted in blue text. We summarize the changes that we made below:

**Experiments**
- Re-preprocessed the FMoW-WildT dataset with a new data split and updated all baseline results in Table 2 in the main paper, Table 19 in Appendix, Figure 1 in Appendix (Eval-Fix), Figure 2 in  in Appendix (Eval-Stream) (Reviewer vzrG).
- Clarified data splits in the Eval-Fix and Eval-Stream settings with two new illustrations in Appendix C.2 (Reviewer QHWk).
- New experiments to investigate the effect of different model backbones in Appendix E.4 (Reviewer QHWk, R2F7).
- Moved mixup results from the Appendix in the initial submission to Section 5.3 of the main paper in the revised version (Reviewer QHWk).
- Added self-supervised learning baselines (SwAV and SimCLR) and an ensemble learning baseline (SWA) under the Eval-Fix setting (Table 2, Table 19 in the Appendix)  (Reviewer QHWk, R2F7, crkT).
- New experiments for temporal invariant learning in Appendix E.3.
  - Investigated the effect of time window length L (Reviewer crkT)
  - Compared baseline performance using overlapping versus non-overlapping time windows (Reviewer rLDr).
- Reduced the number of training examples and analyzed the performance of ERM versus invariant learning and continual learning baselines in Appendix E.5 (Reviewer crkT).

---

**Paper Writing and Relevant Documentation**

- Added more motivation for the temporal distribution shift problem in the Introduction
  - Defined temporal shifts and described real-world scenarios in the Introduction (Reviewer UtTF).
  - Clarified the Eval-Stream setting, which is designed to reflect real-world ML development pipelines (Reviewer vzrG)
- Provided more details of our hyperparameter tuning process in Section 5.1 and Appendix C (Reviewer QHWk).
- Added a maintenance plan in Appendix G discussing how we plan to support the Wild-Time datasets and repository (Reviewers QHWk, R2F7, crkT).
- Put the ethics discussion in a new section (Appendix F.2) and added further discussion (Reviewer UtTF).
- Put the discussion of promising directions in designing temporal robust models in Appendix F.3 (Reviewer crkT).
- Added detailed explanations of why continual learning approaches fail to show improvements in Appendix D (Reviewer QHWk)
- Reorganized tables and figures in both the main paper and the Appendix to make them more readable (Reviewer R2F7, crkT).
- Updated the Github code documentation with more details, including a script for downloading all datasets, detailed instructions for running baseline, and licensing information (Reviewer QHWk, R2F7, crkT).
- Fixed typos in the initial submission (Reviewer rLDr).

Please note that some table and figure indexes have been changed, since we added more results and experiments.

---

### Meta-Review · Area_Chair_foKm · 2022-09-09

**Recommendation:** Accept
**Confidence:** 4

**Metareview:**

I hereby sort the strengths and weaknesses that were mentioned together:

strengths
- the usefulness to the community, and need to develop methods that address OOD data.
- paper well-written

weaknesses
- there are some questions about the datasplit, which seems major, but also addressed
- adding locational shift op top of the temporal shift, which will be a confounding factor in experiments
- there are concerns about the time between certain batches of the data (one reviewer claims it's too big in a certain setting, another mention overlap in another setting)
- information on experimental choices and details and calibration of hyperparameters of the algorithms
- user friendliness (of documentation, Github repo, etc)
Various reviewers have increased their score during the rebuttal.

All together, the reviewers have been very thorough, nicely picking up on these matters. While I appreciate their through job, I want to place the weaknesses into context of the usefulness to the community, that was also consistently mentioned. Some reviewers mention that they are still not 100% convinced, which is an actionable point to the authors: please do keep an open mind and seek to further improve on these points. I see no dealbreakers in the weaknesses, and due to the generally positive sentiment, I am very inclined to give a positive recommendation.

---

### Decision · Program_Chairs · 2022-09-16

Accept